# A gut microbiota rheostat forecasts responsiveness to PD-L1 and VEGF blockade in mesothelioma

Malignant mesothelioma is a rare tumour caused by asbestos exposure that originates mainly from the pleural lining or the peritoneum. Treatment options are limited, and the prognosis is dismal. Although immune checkpoint blockade (ICB) can improve survival outcomes, the determinants of responsiveness remain elusive. Here, we report the outcomes of a multi-centre phase II clinical trial (MiST4, NCT03654833) evaluating atezolizumab and bevacizumab (AtzBev) in patients with relapsed mesothelioma. We also use tumour tissue and gut microbiome sequencing, as well as tumour spatial immunophenotyping to identify factors associated with treatment response. MIST4 met its primary endpoint with 50% 12-week disease control, and the treatment was tolerable. Aneuploidy, notably uniparental disomy (UPD), homologous recombination deficiency (HRD), epithelial-mesenchymal transition and inflammation with CD68[+] monocytes were identified as tumour-intrinsic resistance factors. The log-ratio of gut-resident microbial genera positively correlated with radiological response to AtzBev and CD8[+] T cell infiltration, but was inversely correlated with UPD, HRD and tumour infiltration by CD68[+] monocytes. In summary, a model is proposed in which both intrinsic and extrinsic determinants in mesothelioma cooperate to modify the tumour microenvironment and confer clinical sensitivity to AtzBev. Gut microbiota represent a potentially modifiable factor with potential to improve immunotherapy outcomes for individuals with this cancer of unmet need.

Immune checkpoint blockade (ICB) can achieve clinically meaningful control of mesothelioma in a proportion of patients[1,2]. Single-agent ICB with anti-PD1 inhibitor nivolumab, confers longer survival compared to placebo in patients in the relapsed setting[2], whilst the combination comprising nivolumab and the anti-CTLA4 inhibitor ipilimumab improves survival compared with chemotherapy, particularly in patients harbouring non-epithelioid mesothelioma[1]. However, the cellular and molecular determinants of response remain elusive. Mesotheliomas harbour a low tumour mutation burden of around two mutations per megabase[3,4], but display extensive somatic copy number alterations (SCNAs) including a high clonal 9p21 deletion rate[4] and infiltration of immunosuppressive monocytes[5], both capable of

conferring resistance to ICB[6–8]. Conversely, BRCA-associated protein 1 (*BAP1*), a commonly inactivated tumour suppressor[9,10] has been implicated as a putative sensitiser of ICB, although this has not yet been proven clinically[11].

Gut microbial ecology is a robust predictor of ICB efficacy in multiple cancer types[12,13], but may also be an actionable target to sensitise to ICB[14–16]. To date, the impact of gut microbiota on immunotherapy responses in mesothelioma has not been explored.

The Mesothelioma Stratified Therapy 4 clinical trial (MIST4, clinicaltrials.gov identifier NCT03654833) was designed to examine the efficacy and correlates of response to dual inhibition of programmed death-ligand 1 (PD-L1)-vascular endothelial growth factor (VEGF)

e-mail: df132@leicester.ac.uk

inhibition with atezolizumab and bevacizumab (AtzBev) in patients with pleural or peritoneal mesothelioma who had progressed following first-line chemotherapy. A muti-layer workflow comprising machine learning analysis of next-generation sequenced mesothelioma whole-exome, transcriptome and gut 16S RNA, as well as multiplex immunofluorescence staining of mesotheliomas was conducted; the aim, to infer predictive correlates with potential to rationally advance precision ICB therapy in patients with mesothelioma.

## Results

### Patients

Between January 2020 and June 2021, 30 patients were consented to participate in MIST4, of which 26 were eligible for treatment following screening (Supplementary Fig. 1). The median follow-up time was 16.1 weeks (range, 3.9–49.1). Baseline patient characteristics are summarised in Supplementary Data 1. Median age of the cohort was 68 (IQR 67–74), of which 18/26 (69.2%) were male, 24/26 (92.3%) had pleural mesothelioma, 20/26 (76.9%) had epithelioid histology, 15/26 (57.7%) had lymph node involvement, and 7/26 (26.9%) had metastases. ECOG performance status was 1 in 22/26 (84.6%) patients. The majority of patients, 16/26 (61.5%) reported exposure to asbestos. The complete list of eligibility criteria is summarised in the MIST4 trial protocol in the supplementary materials.

### Efficacy

All 26 patients who were clinically eligible for treatment received at least one cycle. The majority 14/26 (53.8%) had previously received more than one course of systemic therapy (Supplementary Data 2). The median number of cycles received of either drug within 24 weeks

was 4.5 (range 1–8). The median time on study and reasons for discontinuation are shown in Supplementary Data 3.

Disease control rate (DCR) at 12 weeks was assessed radiologically using modified RECIST at a scanning interval of 6 weekly. DCR was achieved in 13/26 patients (50%, 90% CI 32.7–67.3) leading to the rejection of the null hypothesis. Partial response was observed in 1/26 (3.8% 95% CI 0.1–19.6). Progressive disease rate was 8/26 (30.8%, 90% CI 16.3–48.7). Due to the low mRECIST partial response rate, an analysis of extreme phenotypes compared with the mRECIST disease progression subgroup was not statistically feasible. At 24 weeks 7/26 (26.9, 95% CI 11.6–47.8) had disease control (Fig. 1A, B). Radiology was not evaluable in 5/26 (19.2% 90% CI 7.9–36.3) due to clinical progression and symptom burden precluding CT re-evaluation.

### Tolerability and safety

Dose delays occurred in 12/26 (46%) for Atezolizumab and in 9/26 (35%) for Bevacizumab. At least one adverse event (AE) was experienced in 23/26 (88%). Among these 23 patients, the most prevalent adverse event was fatigue in 8/26 (31%), followed by weight loss in 5/26 (19%). AEs were observed in 23/26 (88.5%) of patients with greater than one AE occurring 20/26 (76.9%) of patients (Supplementary Data 4A). All AEs reported (90/130; 69.2%) were grade 1. Treatment-related AEs classed as possibly or probably related to AtzBev occurred in 8/130 of AEs (6.2%) and 9/130 of AEs (6.9%) respectively (Supplementary Data 4B).

Regarding AEs per individual, 14/26 (54%) grade 1 or 2 as their highest grade of AE, 8/26 (31%) had grade 3 and 1/26 (4%) grade 5. The grade 3 AEs involved confusion, vomiting, lower respiratory tract infection, hypertension, joint pain, agitation, bowel obstruction,

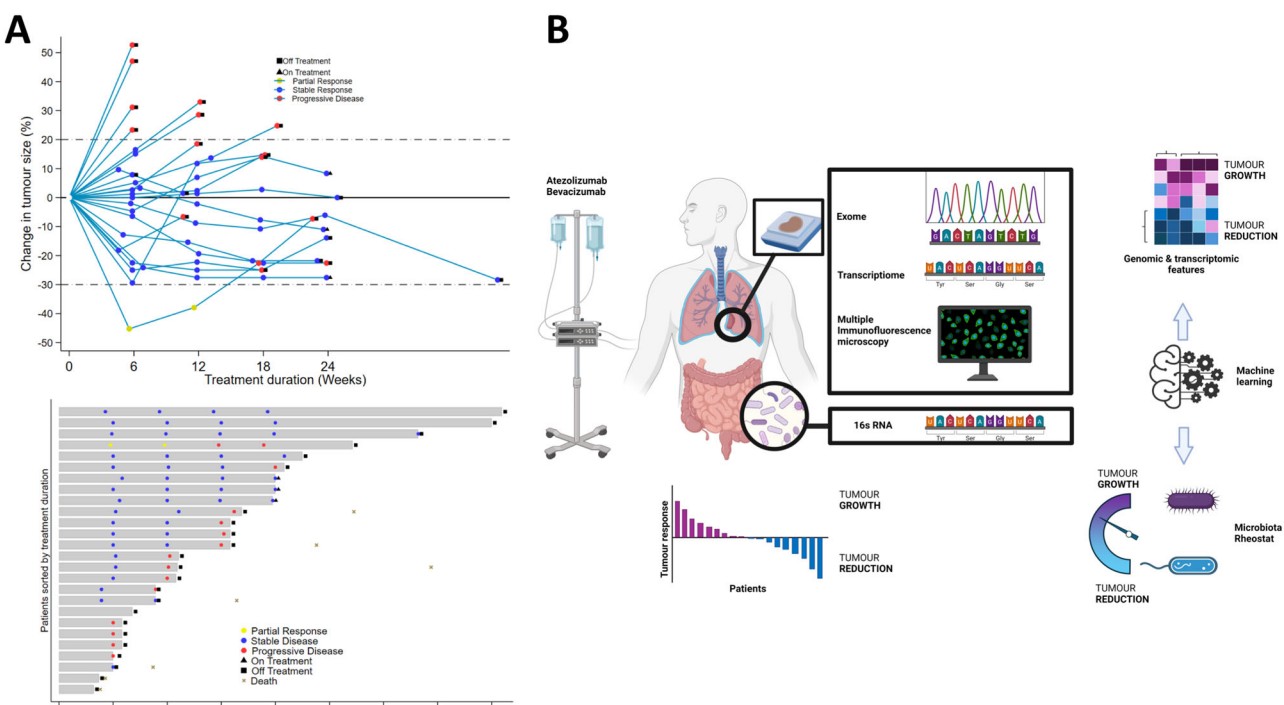

**Fig. 1 | Efficacy and multi-omic analysis workflow in MIST4. A** Upper panel. Spider plot showing the change in tumour size (% assessed by modified RECIST) against time (weeks) within 24 weeks. Dashed lines show threshold corresponding to partial response (−30% lower band) or disease progression (+20% upper band). Lower panel. Swimmer plot showing the duration of treatment measured within 24 weeks. **B** 26 patients with either pleural or peritoneal mesothelioma were recruited into MIST4 and received atezolizumab and bevacizumab (AtzBev), administered intravenously every 21 days until disease progression. Response to

treatment was dichotomised into two groups; showing tumour reduction (blue) or growth (purple) as the best response, respectively. Diagnostic tumour blocks were subjected to whole exome and transcriptome sequencing. Spatial phenotyping by multiplex immunofluorescence was used to interrogate the immune landscape, and gut microbiota was 16S RNA sequenced. Features enriched in tumours showing either growth or reduction were inferred by ensemble machine learning (random forests, extreme gradient boosting).

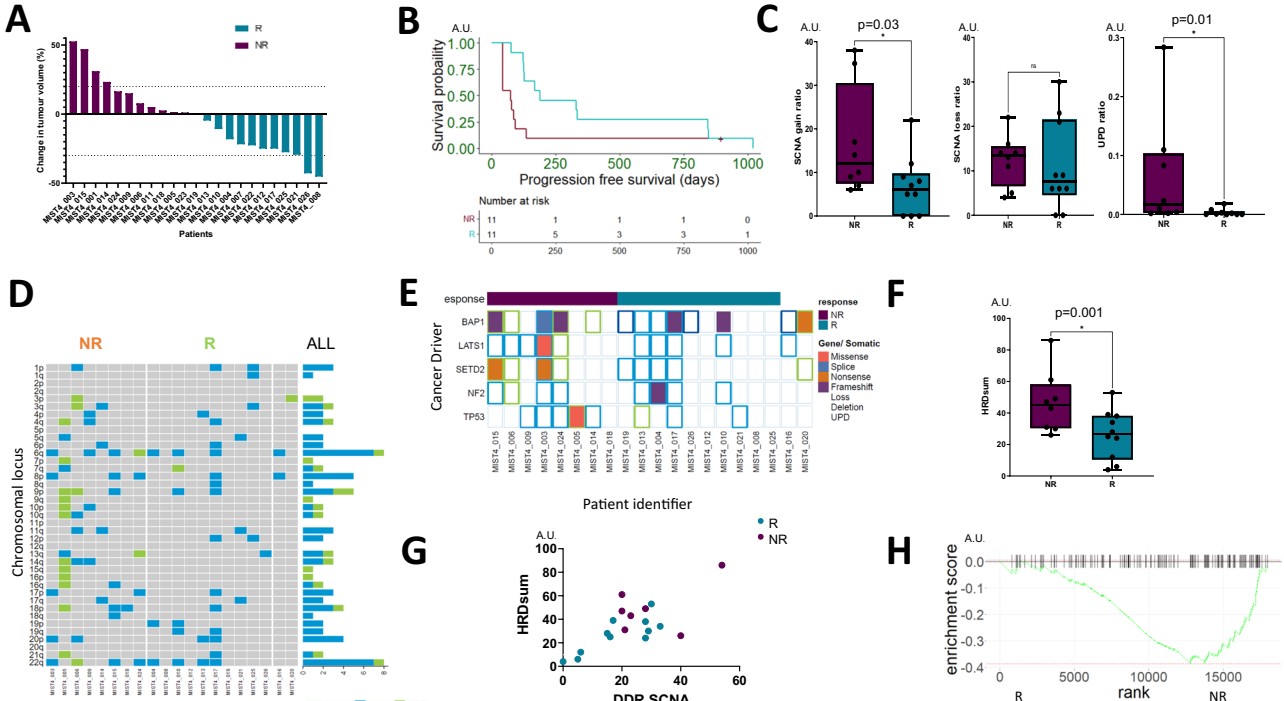

**Fig. 2 | Efficacy and genomic correlates of response to dual PD-L1-VEGF inhibition. A** Waterfall plot showing the best response within 24 weeks for patients enrolled into the MIST4 clinical trial. Response was dichotomised into two equal subgroups; those patients with no tumour reduction (purple, NR) and those with tumour reduction (blue, R). The upper dotted line marks the 20% threshold for progressive disease by modified RECIST 1.1 (mRECIST) and the bottom dotted line, partial response (−30% threshold). **B** R-subgroup patients exhibited longer progression-free and overall survival (PFS and OS respectively) in the R (blue) versus the NR-subgroup (purple). Median PFS was 188 days in the R subgroup and 74 in the NR-subgroup (two-sided Mantel–Cox test $p = 0.02$). **C** Boxplots showing the relative somatic copy number alteration frequency in tumours associated in R-$vs$ NR-subgroup. SCNA gains (left panel; for NR $n = 8$ patients and for R $n = 10$ patients) were higher in NR $vs$ R but not losses (middle panel; for NR $n = 8$ patients and for R $n = 10$ patients), $p = 0.03$ $vs$ 0.56, respectively. Uniparental disomy (UPD) was higher in the R group compared to the NR group (for NR $n = 8$ donors and for R $n = 9$ patients) ($p = 0.01$). Data were analysed with a two-sided Mann–Whitney $U$ test

and data are presented as median value ± IQR. **D** Heatmap showing UPD enrichment in the NR vs R-groups. Cumulative frequency of UPD and SCNA losses are shown in the histogram to the right of the heatmap. **E** Heatmap showing enrichment of tumour suppressor driver alterations involving SETD2, p53 and LATS2 somatic alterations in the NR-(purple) $vs$ R- (blue) subgroup. **F** Boxplot showing enrichment of HRD (HRD$_{sum}$, comprising the sum of LOH, TAI, and LST signatures) in the NR vs the R subgroup (for NR $n = 8$ patients and for R $n = 10$ patients) ($p = 0.001$). Data were analysed with a two-sided Mann–Whitney $U$ test and data are presented as median value ± IQR. **G** SCNAs involving DNA damage response genes (DDR) were positively correlated ($r = 0.73$ $p = 0.03$; a two-sided Mann–Whitney $U$ test) with the combined HRD signature (HRD$_{sum}$), comprising the sum of sub-signatures LOH, TAI, LST. R subgroup is denoted in blue and the NR-subgroup in purple. **H** Gene set enrichment plot showing reduced transcriptional enrichment of DNA repair genes in the NR vs the R subgroup (Benjamini–Hochberg adjusted $p = 0.002$).

dehydration, gastrointestinal bleed, hypophosphatemia, and vena cava obstruction. The grade 5 AE resulting in death occurring in 1/26 (4%) patient was dyspnoea due to mesothelioma (Supplementary Data 5). All AEs (130) are summarised in Supplementary Data 6.

Severe AEs (SAEs) occurred in 11/26 (42%), of which 6/26 (23%) had one SAE, 4/26 (15 %) had two SAEs and 1/26 (4%) had three SAEs (Supplementary Data 7A). SAEs led to permanent treatment discontinuation in 4/26 (15%), with 3/26 (12%) having SAEs that led to permanent treatment discontinuation of both Atezolizumab and Bevacizumab and 1/26 (4%) that led to permanent discontinuation of Bevacizumab alone. SAEs deemed to be related to atezolizumab occurred in 5/26 (19%) and for bevacizumab in 6/26 (23%) (Supplementary Data 7B). A summary of all SAEs is shown in Supplementary Data 8.

### Aneuploidy and HRD predict resistance to AtzBev

To elucidate tumour-intrinsic and extrinsic predictive factors in MIST4, a multi-omic analysis was conducted as summarised in Fig. 1B. germline and mesothelioma DNA were whole exome ($n = 20$, 200× sequencing depth for tumour, 75× for normal DNA with 10 fold coverage >90%) and RNA sequenced ($n = 20$, Supplementary Fig. 2A) using the available archival paraffin-embedded, formalin-fixed diagnostic

tissue blocks. Quality control data related to both DNA and RNA sequencing is provided in Supplementary Data S1–S4. The immune landscape was profiled using 6-colour multiplex immunofluorescence ($n = 18$) and transcriptome deconvolution (Supplementary Data S6–S8), and the gut microbiota by 16S RNA sequencing of the gut microbiota ($n = 17$, Supplementary Data S9–S10).

Radiological response in 22 evaluable patients was dichotomised into equal subgroups exhibiting any tumour reduction (R, shown in blue) $versus$ (vs) no reduction (NR, shown in purple) respectively (Fig. 2A). This classification used throughout the study, encompassed any shrinkage (below or equal to 0% change in tumour volume) as defined by modified RECIST as the R subgroup, and those patients greater than 0% and those patients greater than 0% were defined as NR. Longer progression-free survival corresponding to a follow-up period of 45 months was associated with the R subgroup (188 days) compared with the NR-subgroup (74 days, Mantel–Cox $p = 0.02$, Fig. 2B).

Aneuploidy involving somatic copy number alterations (SCNAs) were enriched in the NR-subgroup (Fig. 2C). This increase was due to higher frequency gains (Mann–Whitney $p = 0.03$, Supplementary Fig. 2B) compared with losses ($p = 0.56$). The NR-subgroup had a higher frequency of UPD, Mann–Whitney $p = 0.01$ Fig. 2C, D, which was exome-wide. This difference was predominantly driven by an outlier

however the distribution was generally wider compared with the R group. *BAP1* protein deficiency was enriched in 5/20 (25%) of NR patients (Fisher exact $p = 0.03$, Supplementary Fig. 3A) but this was not reflected in the proportion of somatic alterations involving *BAP1*. In contrast, p16ink4a protein deficiency (encoded by *CDKN2A*) occurred in 15/20 (75%) patients but was not significantly different between R- *vs* NR-subgroups (Fisher exact $p = 0.99$, Supplementary Fig. 3B). Somatic alterations involving large tumour suppressor kinase 1 (*LATS1*), SET Domain Containing 2, Histone Lysine Methyltransferase (*SETD2*) and tumour protein p53 (*TP53*) were only found in the NR-subgroup (Fig. 2E). Due to the small phase II sample size however, the significance of this observation should be interpreted with caution.

Tumour mutation burden is a well-established and positively correlated predictor of ICB efficacy across multiple cancers[12], and in the MIST4 cohort was 1.1 mutations per megabase (range 0–3.2). Neither the total, non-synonymous or predicted neoantigen burden were differentially enriched in the R- *vs* NR-subgroups (Supplementary Fig. 4). Neoantigens have been postulated to arise in mesothelioma secondary to complex chromosomal rearrangements, the burden of which could correlate with ICB clinical outcomes[17]. However, in the MIST4 cohort, no evidence of differential fusion enrichment (which included a *BAP1-RP11-579D7.1* fusion, Supplementary Fig. 5B) was found in the NR- *vs* R subgroup using either the Arriba or STAR-Fusion packages (Mann–Whitney $p = 0.34$ and $p = 0.74$ respectively, Supplementary Fig. 5A, B). Comprehensive gene re-arrangement analysis using whole genome sequencing was not feasible due to the limited amount of DNA associated with small diagnostic tissue samples.

SCNAs in DNA damage response (DDR) genes ($n = 98$, Supplementary Data 8B) involved homologous recombination (HR) pathway, varied across the cohort ranging from 0 to 54 alterations per patient (Supplementary Fig. 6A). HR deficiency (HRD) was enriched in the NR-subgroup compared with the R subgroup (Mann–Whitney $p = 0.001$) but this was not seen for losses, Mann–Whitney $p = 0.56$ (Fig. 2F, Supplementary Fig. 6B, C). Consistent with this finding, HRD correlated with the burden of DDR SCNAs ($r = 0.73$ $p = 0.03$, Fig. 2G, Supplementary Fig. 6C). Random forest-based machine learning revealed enrichment of *H2AX* and *BRCA2* copy number losses in the NR-subgroup (McNemar's test $p = 0.03$), with evidence of a *BRCA2* germline polymorphism, highlighting a somatic second-hit inactivation. Orthogonal gene set enrichment analysis confirmed DNA repair deficiency at the transcriptional level in the NR-subgroup (Benjamini–Hochberg adjusted $p = 0.002$, Fig. 2H).

### Mesothelioma tumours exhibiting sensitivity to AtzBev are inflamed

Immune-escape mechanisms involving allele-specific loss of human leucocyte antigen (LOHHLA)[18] and PD-L1 tumour proportion score assessed using the 22C3 clone have been observed in mesothelioma[4]. LOHHLA was identified in two patients, one in each of the NR- and R-groups respectively (Fig. 3A). PD-L1 tumour proportion score exceeded 1% of tumour cells in 35% of patients but was not enriched in the R- *vs* the NR-subgroup (Mann–Whitney $p = 0.7$, Fig. 3B).

The R subgroup showed constitutive, transcriptional enrichment of a hallmark inflammatory response gene signature (Benjamini–Hochberg adjusted $p = 0.04$), as well as activation or differentiation of T- and B-lymphocytes (Fig. 3C). No bias involving T-cell or B-cell receptor utilisation was observed (Supplementary Fig. 7A–C). CD8+ effector T-cell enrichment was seen in the R- *vs* NR-subgroup was evidenced by two orthogonal methods; multiplex immunofluorescence microscopy (Mann–Whitney $p = 0.004$) (Fig. 3D) and transcriptome-based immune deconvolution, (Mann–Whitney $p = 0.02$, Supplementary Fig. 8A, B).

Antigen-experienced T-lymphocytes (CD45RO+ CD8+) were more abundant in the R subgroup, (Mann–Whitney $p = 0.03$), and correlated with tumour reduction (Spearman's $r = 0.52$, $p = 0.03$, Fig. 3D, E). Naïve

T-helper lymphocytes (CD45RA CD4+) were also enriched in the R subgroup consistent with a constitutive anti-tumour host immune response (Supplementary Fig. 8C). Transcriptional signatures corresponding to cytokine-cytokine receptor signalling were enriched in the R subgroup, with significantly higher interleukin 18 receptor accessory protein (*IL18RAP*) expression in the R subgroup (Mann–Whitney $p = 0.01$) and monocyte chemotactic protein (*CCL7*) expression in the NR-subgroup (Mann–Whitney $p = 0.02$, Supplementary Fig. 8D–F).

In contrast to the CD8+ T-cell abundance in the R subgroup, monocytes expressing CD68 were enriched in the NR-subgroup (Mann–Whitney $p = 0.04$, Fig. 3F) and were correlated with tumour growth ($r = 0.51$ $p = 0.03$), but inversely associated with CD45RO+CD8+ T-cells (Spearman's $r = -0.61$, $p = 0.007$ Supplementary Fig. 9A). VISTA+ expressing CD68+ monocytes were enriched in the NR-subgroup (Mann–Whitney $p = 0.01$, supplementary fig. 9B).

EMT gene set enrichment in the NR-subgroup determined using single sample hallmark gene set enrichment analysis (GSEA) was associated with shorter progression-free survival in an independent mesothelioma cohort comprising 50 patients (3.5 *vs* 6.1 months, hazard ratio 0.77, Fig. 3G).

### Gut microbiota predict response to AtzBev

Alpha diversity was compared in R versus NR subgroups using 8 methods; Chao1 index, Berger-Parker dominance index, Richness index (Observed-otus), Good's coverage of counts, Pielou's evenness, Shannon index, Simpson's index and Faith's phylogenetic diversity. No statistical difference was observed by the Mann–Whitney test, likely due to the high dimensionality of the data and small sample size (Supplementary Fig. 9D). The R subgroup had a higher type-2 enterotype (enriched for *Provetella*, 33%) compared with the NR group (9%). QIIME2 was used to analyse bacterial composition and random forest feature selection was orthogonally validated by linear discriminant analysis combined with the effect of size measurements (LEfSe). The R subgroup was enriched for the genera *Prevotella* (Mann Whitney test $p = 0.002$), *Butyricioccus* ($p = 0.03$), *Eubacterium ventriosum group* ($p = 0.005$), and *Biophilia* ($p = 0.02$). In contrast, *Erysipeloclostridium* ($p = 0.018$) was enriched in the NR-subgroup (Fig. 4A). The relative phylogenetic distance of these genera is represented in a cladogram in Fig. 4B.

The log ratio of the sum of genera enriched in R- *vs* NR-subgroups respectively, i.e., Log ($G_R/G_{NR}$), was almost 2 logs greater in the R- *vs* NR-subgroup (Mann–Whitney $p < 0.0001$, Fig. 4D). Log ($G_R/G_{NR}$) correlated with radiological response ($r = -0.72$ $p = 0.002$, Fig. 4E), corresponding to an area under the receiver operator curve of 0.94 (95% confidence limits 0.82–0.94 computed using k-fold cross-validation, supplementary Data 8C), outperforming both UPD (0.55) and HRD (0.71, Fig. 4E).

Log($G_R/G_{NR}$) positively correlated with CD8+ T-lymphocytes (Spearman's $r = 0.48$, $p = 0.05$ Fig. 4F), and CD4+T-lymphocytes (Spearman's $r = 0.52$, $p = 0.03$, Supplementary Fig. 9C) but negatively with CD68+ monocyte lineage infiltration (Spearman's $r = -0.41$ $p = 0.05$), Fig. 4F. Log($G_R/G_{NR}$) positively correlated with progression-free survival ($r = 0.47$, $p = 0.03$, Fig. 4G).

Log ($G_R/G_{NR}$) was inversely correlated with both UPD (Spearman's $r = -0.58$ $p = 0.008$) and HRD (Spearman's $r = -0.41$, $p = 0.05$, Fig. 5A). Linear discriminant analysis revealed differential bacterial metabolic profiles as classified by patient response category. The NR- subgroup was enriched for bacterial detoxifying 2-methylcitrate cycle involving odd-chain fatty acid β−oxidation, and allantoin degradation, which yields ammonia as a nitrogen source. In contrast, the more microbially diverse R subgroup showed enrichment of seven metabolic processes comprising gluconate-5-dehydrogenase, phosphodiesterase, R pseudouridine synthase, HAD hydrolase family 1A variant 3, pyridoxal-5-phosphate synthetase, UDP-galactopyranose mutase, and pyrimidine metabolism (Fig. 5B).

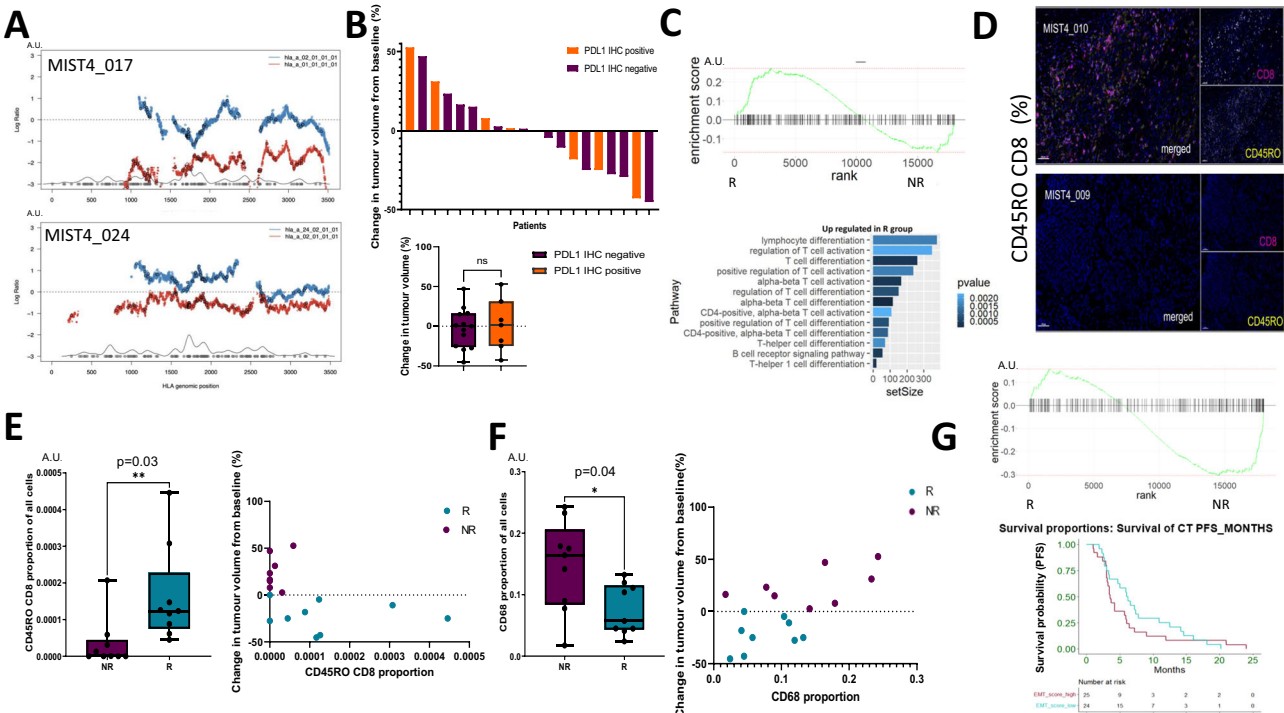

**Fig. 3 | Immune landscape and radiological response in MIST4. A** Histogram showing allele-specific loss of heterozygosity of the human leucocyte antigen relative LOHHLA involving two MIST4 patients (patients MIST4-017 and MIST4-024). **B** Top panel. Waterfall plot showing the expression of PD-L1 (35% overall, orange). Lower panel. Boxplot summarising the PD-L1 expression positive (n = 7 patients) in orange vs negative (n = 13 patients) in purple (p = 0.7). Data were analysed with a two-sided Mann–Whitney U test and data are presented as median value ± IQR. No significant association between PD-L1 TPS and response measured as the change in tumour volume (%) by modified RECIST 1.1. **C** Upper panel. GSEA enrichment plot showing significant enrichment of the hallmark inflammatory response signature in the R- vs NR-subgroup (Benjamini–Hochberg adjusted p = 0.04). Lower panel. Relative activation of immune signatures in R- vs NR- subgroups supporting constitutive T-cell activation in the R subgroup. **D** Immunofluorescence microscopy comparing the proportion of antigen-experienced CD45RO⁺ CD8⁺T-lymphocytes in an exemplary R-group patient (MIST4-010) with an NR-subgroup patient (MIST4-09). Bar represents 50 μM. **E** Left panel. Boxplot showing greater proportion of CD45RO⁺ CD8⁺ effector T-cells in the R- vs NR-subgroup (for NR n = 9 patients and for R n = 9 patients p = 0.03); the

boxplot is bounded by the 25TH/75TH percentiles, showing the median (horizontal line), maximum (upper whisker) and minimum (lower whisker). Data were analysed with a two-sided Mann–Whitney U test and data are presented as median value ± IQR. Right panel. Scatter plot showing a correlation between CD45RO⁺ CD8⁺ effector T-cells and tumour reduction (r = −0.52, p = 0.03). Data were analysed with a two-sided Spearman r test. **F** Left panel. Boxplot showing greater proportion of CD68⁺ myeloid cells in the R- vs NR-subgroup (for NR n = 9 patients and for R n = 9 patients p = 0.04; the boxplot is bounded by the 25th/75th percentiles, showing the median (horizontal line), maximum (upper whisker) and minimum (lower whisker)). Data were analysed with a two-sided Mann–Whitney U and data are presented as median value ± IQR. Right panel. Scatter plot showing the correlation between CD68⁺ monocytes with tumour growth (r = 0.51, p = 0.03; two-sided Spearman r test.). Blue points correspond to the R subgroup and purple to the NR-subgroup. **G** Top panel. Gene set enrichment plot showing EMT enrichment in the NR- sub-group. Lower panel. Kaplan–Meier curves showing shorter overall survival (3.5 for high EMT vs 6.1 months, HR 0.77) for patients exhibiting EMT enrichment signature in an independent cohort.

## Discussion

ICB improves the survival outcome for patients with mesothelioma[2,19], however only a minority will experience a radiological response. VEGF confers an immunosuppressive microenvironment, biasing M2 to M1 macrophage polarity and reduced tumour infiltration of CD8⁺ and CD4⁺ T-lymphocytes[20]. In MIST4, VEGF blockade was employed to augment ICB. AtzBev combined with chemotherapy[21] is effective in non-small cell lung cancer and is currently being explored in a ran-domised phase III trial in the frontline setting in patients with meso-thelioma (BEAT MESO, Trials.gov ID NCT03762018).

MIST4 met its primary endpoint, however, half of the patients progressed on therapy by 12 weeks. The mechanisms underpinning the response to ICB in mesothelioma remain elusive. To address this, first, we examined tumour-intrinsic factors that might regulate ICB efficacy. Tumour mutation burden, an established predictor of ICB efficacy in other cancers[7], was not associated with clinical outcome in MIST4; neither the load of non-synonymous mutations or neoantigens.

We identified signatures of HRD in MIST4 that were enriched in patients with treatment-refractory mesothelioma. HRD was asso-ciated with the enrichment of SCNAs involving DNA damage

response regulator *BRCA2*. HRD in mesothelioma likely underpins the observed response to PARP inhibition[22] currently being explored in a randomised phase II clinical trial in patients with relapsed mesothe-lioma (NERO trials.gov identifier NCT05455424). The enrichment of HRD in the NR group warrants the exploration of combined ICB and PARP inhibition in patients with platinum-sensitive mesothelioma, which is being explored in the MIST5 trial which is now fully recrui-ted. The UNITO-001 phase 2 trial[23] showed potential tumour activity in patients with mesotheliomas harbouring germline *BAP1* and *BRCA2* mutations. It should be noted that in the PR505 clinical trial, HRD was reported to be predictive of response[24]. However, ICB was coadmi-nistered with a platinum doublet which is known to be sensitised to HRD. In a case report of an exceptional responder to ICB, HRD was also enriched[25]; however, this patient had marked tumour inflam-mation, which is likely to have been critical to underpinning the dramatic immune response.

Mesotheliomas harbour extensive genomic rearrangements with neoantigen-generating potential that could influence the efficacy of ICB[17], however, we found no evidence of an association between fusion burden and response in MIST4. Furthermore, inferred B- and T-cell

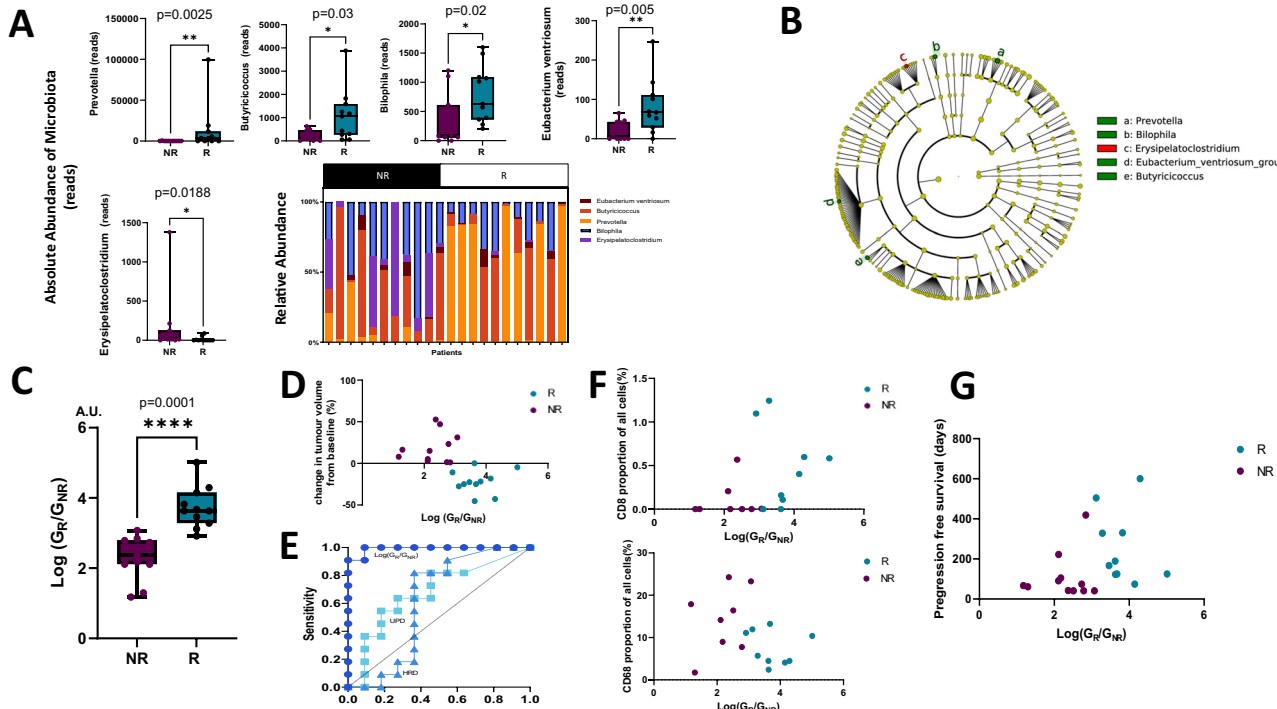

**Fig. 4 | Gut microbiota ratio of R- vs NR-subgroups predicts response and shapes the immune microenvironment. A** Boxplots showing the comparative proportion of genera in the two subgroups (*Prevotella* ($p = 0.002$), B*utyricioccus* ($p = 0.03$), *Eubaterium ventriosum* ($p = 0.005$), *biophilia* ($p = 0.02$), *erysipeloclostridium* ($p = 0.018$)). For each analysis, $n = 11$ patients from the NR-subgroup and $n = 11$ patients from the R subgroup have been used. Data were analysed with a two-sided Mann–Whitney U test and data are presented as median value ± IQR. The boxplots are bounded by the 25th/75th percentiles, showing the median (horizontal line), maximum (upper whisker) and minimum (lower whisker). **B** Microbiota cladogram showing the phylogenetic distance of R- vs NR-subgroup enriched genera. **C** Boxplot showing a significantly higher logarithmic ratio of enriched gut microbial genera, log ($G_R/G_{NR}$) in the R subgroup (blue) compared with the NR-subgroup (purple), $p < 0.0001$. For each analysis, $n = 11$ patients from the NR-subgroup and $n = 11$ patients from the R subgroup have been used. Data were analysed with a two-sided Mann–Whitney U test and data are presented as median value ± IQR. The boxplot is bounded by the 25th/75th percentiles, showing the median (horizontal line), maximum (upper whisker) and minimum (lower whisker). **D** Scatter plot showing the correlation between log($G_R/G_{NR}$) and radiological response ($r = -0.72$ $p = 0.002$). Blue points correspond to the R subgroup and purple to the NR-subgroup. Data were analysed with a two-sided Spearman $r$ test. **E** Receiver operator curve corresponding to log($G_R/G_{NR}$) versus response. Area under the ROC curve is 0.94 computed with k-fold cross-validation (95% confidence limits 0.82–0.94). ROC curves are superimposed for UPD and HRD (AUC. **F** Top panel. Scatter plot showing a positive correlation between log($G_R/G_{NR}$) and CD8⁺ T-lymphocytes ($r = 0.48$, $p = 0.05$). Data were analysed with a two-sided Spearman $r$ test). Lower panel. Scatter plot showing a negative correlation between log($G_R/G_{NR}$) and CD68⁺ monocytes ($r = -0.38$ $p = 0.13$). Data were analysed with a two-sided Spearman $r$ test. Blue points correspond to the R subgroup and purple to the NR-subgroup. **G** Scatter plot showing a positive correlation between log($G_R/G_{NR}$) and progression-free survival ($r = 0.47$, $p = 0.034$). Data were analysed with a two-sided Spearman $r$ test. Blue points correspond to the R subgroup and purple to the NR-subgroup.

clonality failed to identify any bias in clonal expansion between responders and treatment-refractory patients.

Mesotheliomas exhibit extensive aneuploidy and are driven by copy number alterations affecting a restricted number of tumour suppressors, which include loss of 3p21 (harbouring *BAP1*) and 9p21 (*CDKN2A*). The latter has been reported to be associated with resistance to ICB[6] in a pan-cancer study, and its co-deletion with *MTAP* is associated with defective T-cell function[26], however, *CDKN2A* was not predictive in MIST4. Contrary to reports implicating *BAP1* as a putative ICB sensitiser, we observed *BAP1* inactivation detected by either loss of nuclear localisation or expression, to be enriched in the treatment-refractory group, which also harboured enrichment of *SETD2* and *TPS3*.

Arm and chromosomal-level aneuploidy have been reported to be associated with reduced cytotoxic immune infiltration, M1:M2 TAM ratio and to confer resistance to ICBs[8]. We observed extensive chromosome-level aneuploidy in MIST4 with a bias involving uniparental disomy and SCNA gains rather than losses in the treatment-refractory group.

The tumour immune microenvironment is a critical regulator of response to ICB. As previously reported in the CONFIRM phase III trial[2], we found that tumour expression of PD-L1 was not associated with treatment outcome in MIST4. However, CD68-expressing monocytes

which account for around 20-30% of infiltrating cells[5] was enriched in treatment-refractory mesotheliomas. CD68⁺ monocytes exclude CD8⁺ T-cells from the tumour immune microenvironment in a colony-stimulating factor 1 receptor-dependent manner[27] and could explain this reciprocal relationship. TAMs also promote EMT[28] which we and others[29] have observed in response to dual PD-L1-VEGF inhibition. It should be noted that our 6-colour multiplex immunofluorescence analysis was limited in capturing only a proportion of the mesothelioma immune landscape. With accelerating advances in high-plex imaging, deeper analysis of immunophenotype and its correlation with ICB response will become feasible.

Gut microbiota have emerged in recent years as a significant, tumour extrinsic factor associated with sensitivity to ICB[12,13]. Preclinical and clinical studies have identified specific gut bacteria as regulators of the tumour microenvironment in diverse cancers including lung and melanoma[12,13]. Using machine learning-based inference, we identified differentially genera in R vs NR subgroups. The log ratio of these genera behaved as a rheostat, suggesting that diet-related ecologies could underpin a reciprocal abundance pattern and regulate response to ICB[30,31].

The mechanism by which different gut microbiota could reciprocally regulate the mesothelioma tumour microenvironment and

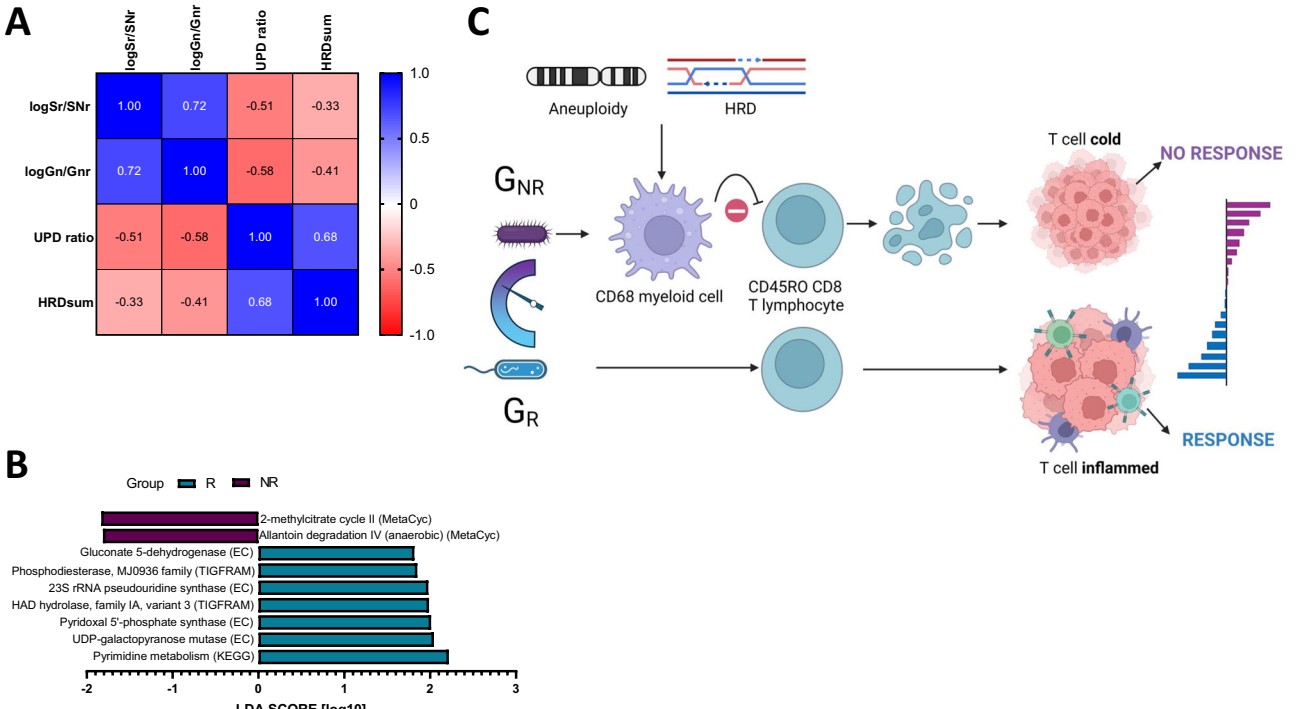

**Fig. 5 | Gut microbiota correlates with genomic instability and exhibits AtzBev response-specific metabolic profiles. A** Correlation matrix showing negative correlations between Log $(G_R/G_{NR})$ and UPD ($r = -0.58$, $p = 0.008$), HRD$_{sum}$ ($r = -0.41$, $p = 0.05$) respectively. UPD and HRD$_{sum}$ were positively correlated ($r = 0.68$, $p = 0.003$). Data were analysed with a two-sided Spearman $r$ test. Predictive metagenomic analysis identifies differential microbial pathway enrichment in R- $vs$ NR patients in MIST4. **B** Histogram showing the linear discriminant analysis related metabolic process enrichment in the NR- $vs$ the R subgroup. **C** Model to illustrate a possible link between the gut microbiota rheostat log ($G_R/G_{NR}$) and mesothelioma response in MIST4.

response to ICB in mesothelioma is unknown. It is possible that both tumour-intrinsic and extrinsic factors are co-variate with response but not causally linked. *IL18RAP* was found to be positively correlated with response and is an accessory subunit of the heterodimeric IL18 receptor which can stimulate memory (but not naïve) T-lymphocytes and can facilitate IFNγ signals downstream of the NLRP1 and NLRP3 inflammasome[32]. Elevated *IL18RAP* expression in treatment-sensitive patients was associated with increased IFNγ signalling, which could potentially couple gut microbial composition to ICB sensitivity. Although specific gut microbiota biochemical pathways were shown to be enriched in the R group, the causal associations with ICB response remain unknown.

We propose a model (Fig. 5C) in which the relative ratio of specific bacterial genera constitutes a rheostat, that shapes the immune microenvironment, inversely correlating with CD68[+] myeloid cell infiltration to likely enable efficient CD8[+] T-cell infiltration, activation, and clinical tumour suppression. Gut microbiome sequencing in both the recruited MIST3 (PD1-AXL) and MIST5 (PD1-PARP) phase II trials may provide further evidence to reinforce this model.

In summary, gut bacteria composition forecasts the sensitivity to immunotherapy in patients with mesothelioma, highlighting a potentially modifiable strategy through dietary modification, such as a high-fibre diet[14] to improve clinical outcomes.

## Methods
### Study design and participants
MIST4 is a multi-centre, one-stage, single-arm open-label phase II study at three UK centres; the University Hospitals of Leicester National Health Service Trust, and the Northern Centre for Cancer Care, Newcastle upon Tyne, and the University Hospital Southampton NHS Trust.

Patients were eligible for the study if they were aged over 18, had evidence of radiologically progressing, histologically confirmed malignant mesothelioma after at least one course of systemic treatment for mesothelioma that included standard first-line pemetrexed and either cisplatin or carboplatin. Patients could be enrolled irrespective of the histological subtype and localisation of their primary mesothelioma - *i.e.*, pleural, peritoneal, or other. Any line of treatment was permitted (excluding any prior to immunotherapy) with prior therapy completing no less than 14 days before treatment was initiated.

Patients were required to have measurable disease by modified Response Evaluation Criteria in Solid Tumours for malignant mesothelioma (mRECIST1.1), predicted life expectancy of 12 weeks or more, Eastern Cooperative Oncology Group performance status score of 0–1, adequate haematological (full blood count including total white cell count, neutrophils, platelets and haemoglobin), renal (urea and electrolytes), and liver function tests (including bilirubin, alkaline phosphatase, alanine transaminase) and willingness to undertake research blood tests and optional tissue re-biopsy for translational research (please refer to the protocol in the supplementary materials).

Exclusion criteria included diagnosis or treatment of any other cancer within the 5 years before study entry, treatment with any agent with no marketing authorisation within 30 days before study entry, and palliative radiotherapy in the 4 weeks before baseline computerised tomography (CT) scan, uncontrolled brain metastases, and cardiac, respiratory, hepatic or renal insufficiency (full exclusion criteria are given in the supplementary information). The protocol was approved by the East Midlands Leicester South Research Ethics Committee (reference 18/EM/0118), and the Medicines and Healthcare Products Regulatory Agency (MHRA). Additional tissue analysis was conducted under research ethics approval 14/LO/1527, a translational research platform entitled Predicting Drug and Radiation Sensitivity in Thoracic Cancers. The study was further approved by the University Hospitals of Leicester NHS Trust (reference IRAS131283 and 14/EM/1159) with the University of Leicester being a sponsor.

The study was completed in accordance with the provisions of the Declaration of Helsinki and Good Clinical Practice guidelines as defined by the International Conference on Harmonisation. Written informed consent was obtained from all patients before enrolment.

## Procedures

In stage 1 (MiST Master protocol), patients were eligible if they had progressed following at least one course of prior systemic treatment for mesothelioma that included standard first-line pemetrexed and either cisplatin or carboplatin. The initial study doses of Atezolizumab and Bevacizumab (Roche, Basel, Switzerland) were 1200 mg intravenously and 15 mg/kg iv in 21-day cycles for a period of 24 weeks. Response was assessed by CT scan every 6 weeks until week 24; thereafter, CT scans were done every 12 weeks. CT scans were assessed by mRECIST1.1, and the radiological review was masked. Tumour response assessments were based on a local review (ie. no central review of response data). Patients who had disease control beyond 24 weeks could continue to receive atezolizumab and bevacizumab off-trial until disease progression, unacceptable toxicity, or withdrawal of consent. Any patient not receiving a single dose of the study drug was removed from the efficacy population and replaced until all 26 patients were recruited (please refer to the full protocol in the supplementary materials).

Patients had safety monitoring visits (including physical examination, blood tests, and toxicity assessments) at the end of each cycle, with additional visits at day 15 of cycles one and two, and follow-up visits 30 days and 6 months after the last dose (Supplementary Information). Dose interruption was allowed for National Cancer Institute Common Terminology Criteria for Adverse Events (NCI CTCAE; version 4.03) grade 3 or 4 toxicity, for a maximum of 14 days, until complete recovery or reversion to grade 2 toxicity. Up to three dose reductions were allowed; corresponding to 150 mg (dose level −1), 100 mg (dose level −2), or 50 mg (dose level −3), twice a day. Dose escalations were not permitted.

## Outcomes

The primary endpoint was disease control rate at 12 weeks, defined as the number of patients with complete response, partial response, or stable disease, as a proportion of the total number of patients who received at least one dose of the study drug. The 12-week landmark was used to establish a threshold for activity and has been frequently used as a primary endpoint in phase 2 studies in mesothelioma[9]. There is currently no standard of care in the relapsed disease setting. In the placebo group of the negative Vantage phase 3 trial (comprising 332 patients)[10] median progression-free survival was 6 weeks. Consequently, we estimated that a 12-week disease control rate of 50% would approximate to a doubling of expected progression-free survival (for placebo), indicating a potentially useful treatment. Secondary endpoints were the safety and toxicity profile, disease control rate at 24 weeks, and best objective response rate. Objective response rate was defined as the proportion of patients whose best overall response was complete or partial. Safety was assessed by the incidence of adverse events, reported according to the NCI CTCAE, version 4.03.

## MIST4 statistical analysis

We used a single-stage A'Hern design with a type 1 error rate (one-sided) of 0.05 and power of 80%. The 12-week disease control rate parameters were set at $p_o = 0.25$ (i.e., a true disease control rate of 25% at 12 weeks would be too low, requiring no further evaluation therefore accepting the null hypothesis) and $p_1 = 0.50$ (i.e., a true disease control rate of 50% at 12 weeks would be sufficient to warrant further evaluation). These parameters required a total of 26 evaluable patients to be analysed. On the basis of these assumptions, if 11 or more of the 26 enrolled patients achieved disease control at 12 weeks, we would conclude that the criteria for success had been met. The efficacy population was defined as all patients who received at least one dose of the study drug. The primary outcome was analysed in the efficacy population; we calculated the disease control rate at 12 weeks with exact two-sided 95% confidence intervals (CIs). All secondary endpoints and safety outcomes were analysed in the efficacy population. The disease control rate at 24 weeks and objective response rate with exact (two-sided) 95% CIs were calculated. Serious adverse events and adverse events were summarised by number, event, frequency, outcome, treatment given, severity (grade), and investigator-assessed relatedness to atezolizumab and bevacizumab.

Safety and toxicity outcomes were determined for the safety population, defined as all participants who received at least one dose of trial medication. Primary and secondary analyses were done after all patients completed 24 weeks of treatment, or at withdrawal. Safety reports of patients who were on treatment beyond 24 weeks, up to 6 months, are provided to the drug provider separately. Categorical variables were summarised by frequencies and continuous variables were summarised by medians with interquartile ranges (IQRs). Chi-square ($\chi^2$) test or Fisher's exact test (in case of low event rates [$n < 5$]) were used to investigate the association between BRCA1 and *BAP1*, p16ink4a, and PD-L1 expressions and response outcome.

In a post-hoc analysis (not protocol specified), progression-free survival was measured in weeks from the first dose to the date of progressive disease or death from any cause, censoring patients at the last known study visit assessment without evidence of disease or death. Overall survival was measured in weeks from the first dose to the date of death from any cause, censoring all other patients at data lock. Median progression-free and overall survival were estimated using the Kaplan−Meier method, and for comparison of survival curves Mantel−Cox *p*-value < 0.05 was considered significant. MIST4 statistical analyses were performed using STATA version 16.0.

## Formalin-fixed paraffin-embedded (FFPE) tissue assessment and processing

FFPE tissue biopsy blocks were used for nucleic acid (DNA and RNA) extraction. After sectioning haematoxylin and eosin (H&E) stained slides were examined by a histopathology advanced biomedical scientist with support from a consultant histopathologist, who identified and marked representative areas of tumour on the H&E slides. Multiple tissue cores (1.0 mm each in size) were taken from the marked areas.

DNA and RNA were isolated from these tissue cores using the MagMAX™ FFPE DNA/RNA Ultra Kit (ThermoFisher Scientific, Waltham, MA, USA #A31881) on the Kingfisher™ Flex sample purification system (ThermoFisher Scientific, Waltham, MA, USA) as per manufacturer's instructions. DNA was quantified using the Qubit™ 1× dsDNA HS assay (ThermoFisher Scientific, Waltham, MA, USA #Q33230) and RNA was quantified using the Qubit™ RNA HS assay kit (ThermoFisher Scientific, Waltham, MA, USA Q32852) on the Qubit™ 4.0 fluorometer (ThermoFisher Scientific, Waltham, MA, USA) according to manufacturer's instructions.

## Whole-exome sequencing

A total amount of 1.0 μg genomic DNA per sample was used as input material for the DNA sample preparation. Germline DNA was isolated from buffy coat using the QIAamp DNA Blood Mini Kit (50) (Qiagen 51104). DNA was quantified using the Qubit™ 1 x dsDNA HS assay (ThermoFisher Scientific #Q33230) on the Qubit™ 4.0 fluorometer. Sequencing libraries were generated using the Agilent SureSelect Human All ExonV6 kit (Agilent Technologies, San Diego, CA) following the manufacturer's recommendations and index codes were added to each sample. Briefly, fragmentation was carried out by a hydrodynamic shearing system (Covaris, Massachusetts, USA) to generate 180−280 bp fragments. Remaining overhangs were converted into blunt ends via exonuclease/polymerase activities and enzymes were removed. After adenylation of 3′ ends of DNA fragments, adaptor

oligonucleotides were ligated. DNA fragments with ligated adaptor molecules on both ends were selectively enriched in a PCR reaction. After PCR reaction, library hybridize with Liquid phase with biotin labelled probe, after which streptomycin-coated magnetic beads are used to capture the exons of genes. Captured libraries were enriched in a PCR reaction to add index tags to prepare for hybridization. Products were purified using AMPure XP system (Beckman Coulter, Beverly, USA) and quantified using the Agilent high-sensitivity DNA assay on the Agilent Bioanalyzer 2100 system. Qualified exome capture libraries were then sequenced on the Illumina NovaSeq 6000 platform, according to standard protocols, for 150 bp paired-end multiplexed sequence.

## Processing of WES sequencing data

After removing sequencing reads with low quality and adaptor bases using FASTP, clean reads were aligned to human reference genome (UCSC hg19) using Burrows-Wheeler Aligner (bwa-0.7.17). Mapped genomes were sorted using Sambamba (v0.6.7). Duplicate reads were marked using Picard tools.

(v2.18.9). Somatic SNVs and INDELs were detected with VarScan2 and MuTect2[2] jointly. Briefly, VarScan2 somatic (v2.3) were used to do somatic variants calling with default parameters, except for the following: minimum coverage for normal and tumour sample were set to 10 and 8 separately, minimum variant frequency was adjusted to 0.01 and tumour purity was set to 0.5. As to MuTect2 dealing process, we used MuTect2 contained in GATK bundle (4.0.5.1), with default parameter. ANNOVAR was used for functional annotation of variants.

## SNV and INDEL filtering

To reduce false positive variant calls, further filtering strategies were used on the mutation detection results of both MuTect2 and VarScan2. A variant was retained when it was both detected by MuTect2 and VarScan2 (somatic p-value ≤ 0.1 for SNV and ≤0.05 for INDEL) with vaf >2%, or only detected by VarScan2 with a VAF > 5%. In matched tumour sequencing data, the VAF for the variant should be <1% and reads number for alternative alleles was less than 5 for SNV or 2 for INDEL. Besides, variants located on the regions of simple repeats and segmental duplications were also removed. The population frequency of the SNV did not exceed 1% in any of the following population based database – 1000 Genome, EXAC or ESP6500. An additional filter was applied to exclude artefact mutations introduced by the preparation of FFPE specimens and sequencing libraries, which are characterized as the bias of the variant read support by DKFZBiasFilter.

## SCNA calling, tumour purity and ploidy estimation

We use ASCAT to estimate somatic copy number alternations (SCNA) in paired tumour tissue/normal tissue sequencing datasets. The estimated tumour purities and ploidies were also corrected by manual review of ABSOLUTE results. Allele counts of positions from 1000 genomes were generated using AlleleCounter, and minimum coverage of 20 for normal sample was used for filtration. LogR and BAF values were produced for each region and concatenated into one matrix separately for each patient. LogR values were subsequently corrected using a GC wave correction implemented in ASCAT, and only heterozygous BAF values were reserved for further analysis. Allele-specific segmentation was performed to generate segmented logR and BAF data by ascat.aspcf. Manual verification was used to select the optimal model for ploidy and cellularity using an orthogonal measure based on ABSOLUTE results and mutation variant allele fraction. And then ASCAT was re-run to obtain the final allele-specific copy number data using reviewed cellularity and ploidy.

## HRD score calculation

Homologous recombination deficiency (HRD) scores are determined using the scarHRD R package. HRD score based on allele-specific copy numbers is sum of loss off heterozygosity (LOH), telomeric allelic imbalance (TAI), large-scale transitions (LST) scores. HRD-LOH score is the number of 15 Mb exceeding LOH regions which do not cover the whole chromosome. HRD-TAI is allelic imbalances that extend to the telomeric end of a chromosome. HRD-LST is defined as chromosomal break between adjacent regions of at least 10 Mb, with a distance between them not larger than 3 Mb.

## Clonality analysis

A modified version of PyClone was used to estimate the cancer cell fraction (CCF) of the mutations and perform clustering analysis. For a given mutation we first calculated the observed mutation copy number, nmut, describing the fraction of tumour cells carrying a given mutation multiplied by the number of chromosomal copies at that locus using the following formula (1):

$$nn_{mut} = VAF\frac{1}{p}\left[pCN_t + CN_n(1-p)\right] \tag{1}$$

where VAF corresponds to the variant allele frequency at the mutated base, and $p$, $CN_t$, $CN_n$ are respectively the tumour purity, the tumour locus specific copy number, and the normal locus specific copy number ($CN_n$ was assumed to be 2 for autosomal chromosomes). We then calculated the expected mutation copy number, nchr, using the VAF and assigning a mutation to one of the possible local copy numbers states using maximum likelihood. In this case only the integer copy numbers were considered. Mutations were then clustered using the PyClone Dirichlet process clustering. For each mutation, the observed variant count was used and reference count was set such that the VAF was equal to half the pre-clustering CCF. Given that copy number and purity had already been corrected, we set the major allele copy numbers to 2 and minor allele copy numbers to 0 and purity to 0.5; allowing clustering to simply group clonal and subclonal mutations based on their pre-clustering CCF estimates. We ran PyClone with 10,000 iterations and a burn-in of 1000, and default parameters, with the exception of --var_prior set to 'BB' and --ref_prior set to 'normal'.

## HLA typing and HLALOH

HLA typing for MHC class-I genes was carried out using POLYSOLVER(v1.0) software for all 28 normal samples' bam files, with default parameters. In brief, reads in the WES data potentially originate from HLA gene region were extracted out and then aligned to genomic sequence library of all known HLA alleles based on IMGT, using Novoalign packaged in POLYSOLVER. After which, a two-step Bayesian classification approach was used to infer the two alleles for each HLA class-I genes (HLA-A, HLA-B and HLA-C). A crucial part of neoantigen presentation is the HLA class-I gene products, which can present tumour associated epitopes to T-cells and then trigger an adaptive immune response. Loss of heterozygosity in HLA genes may lead to decreased ability to present productive tumour neoantigens, which could facilitate immune evasion of cancer. LOHHLA software was used to evaluate HLA loss for all 118 tumour samples, based on the alignment results of both tumour and corresponding normal samples, inferred tumour purity and ploidy information, and the HLA class-I genotyping results detected above. In brief, HLA reads were extracted and re-aligned to the patient-specific HLA-I alleles, then HLA gene specific log ratio was calculated based on coverage information on mismatch positions between homologous HLA alleles, and finally, HLA haplotype specific copy number was determined. In the analysis, items with PVal_unique ≤0.01 (difference in log ratio between allele 1 and allele 2 ≤ 0.01) were considered as a LOH event.

## Neoantigen prediction

In this analysis, neoantigens were defined as 8–11-mer peptides resulted from somatic SNVs or InDels which lead to amino-acid changes

and, binding affinity score between remodelled peptide and respective patient's HLA class-I molecules was <500 nM. Somatic mutation VCF files both from VarScan2 and Mutect2 were annotated by Variant Effect Predictor (Version 84) with default parameter, except for the using of 'downstream' and 'wild-type' plugins offer by pVACseq53. After annotation, the variants items lead to peptide changes were extracted out for downstream analysis. Bam-readcount (0.8.0) was used to acquire sequencing-based read depth information on each selected variant for both tumour and matched normal samples. Annotated non-synonymous mutations, sequencing-based information as well as HLA class-I gene typing results inferred by POLYSOLVER were feed into pVACseq(4.0.9) for neoantigen pre- diction. For each pVACseq run, epitope prediction was done by both NetMHC and NetMHCpan algorithms packed in pVACseq toolkit, epitope length was set to 8–11 and tumour DNA VAF cut-off was set to 10, with default parameters used for all other settings. Epitope prediction was performed based on the selected prediction algorithms, after which, sequencing-based information was integrated to enable filtering of neoantigen candidates (Normal Coverage ≥5×, Normal VAF ≤ 2%, Tumour Coverage ≥10×, Tumour VAF ≥10%). Inferred neoantigen candidates were selected out and those with binding affinity fold change >2 were considered with higher priority level, which means the ratio of binding affinity score between wild-type peptide and mutated peptide. The greater this value, the stronger of the binding affinity after mutation compared with wild-type epitope.

## RNA-sequencing library construction and sequencing

Total RNA from cores of formalin-fixed, paraffin-embedded (FFPE) tissue was isolated using a MagMAX™ FFPE DNA/RNA Ultra Kit (ThermoFisher Scientific, Waltham, MA, USA #A31881) on the King-fisher™ Flex sample purification system (ThermoFisher Scientific, Waltham, MA, USA) as per manufacturer's instructions. and RNA integrity was assessed using the Bioanalyzer 2100 system (Agilent Technologies, CA, USA). A total amount of up to 2 RNA per sample was used as input material for the RNA sample preparations. Sequencing libraries were generated using NEBNext® UltraTM RNA Library Prep Kit for Illumina® (NEB, USA) following manufacturer's recommendations and index codes were added to attribute sequences to each sample. Briefly, mRNA was purified from total RNA using poly-T oligo-attached magnetic beads. After fragmentation, the first strand cDNA was synthesized using random hexamer primer followed by the second strand cDNA synthesis using dTTP. Remaining overhangs were converted into blunt ends via exonuclease/polymerase activities. After adenylation of 3' ends of DNA fragments, NEBNext Adaptor with hairpin loop structure were ligated to prepare for hybridization. In order to select cDNA fragments of preferentially 150-200 bp in length, the library fragments were purified with AMPure XP system (Beckman Coulter, Beverly, USA). The concentration of each library was measured with real-time PCR. Pools of the indexed library were then prepared for cluster generation and PE150 sequencing on Illumina NovaSeq 6000.

## RNA-sequencing analysis

Fastp (0.12.2) was used to remove low-quality reads and reads containing sequencing adapters. RSeQC (v5.0.3) was employed to perform quality assessment of RNA sequencing. The processed reads were aligned using STAR (2.6.1d) onto the human genome reference (UCSC hg19), and the transcripts were annotated based on gencode V19 gene models. Only the reads unique to one gene and which corresponded exactly to one gene structure were assigned to the corresponding genes by using HTSeq. Counts were normalized for library size using estimateSizeFactors in Deseq2. FPKM data were generated using the fpkm function in Deseq2. STAR-Fusion (1.9.0)[15] was applied to predict gene fusion events from RNA-seq data.

## Immune repertoire analysis

We applied TRUST4 (v1.0.0) to obtain TCR and BCR clonotypes from bulk-RNA-seq data for each sample. Raw pair-end reads were aligned to hg19 TCR/BCR sequences, and candidate reads were extracted to perform de novo assembly on V, J, and C genes including the hyper-variable complementarity determining region 3 (CDR3). The assembled consensus sequences were re-aligned to IMGT reference gene sequences for annotation. The statistics of TCR and BCR, including abundance, richness, Shannon entropy and clonality, were compared between reduction and no-reduction groups using the Kruskal–Wallis rank-sum test.

## Immune deconvolution

Pre-processing of raw RNA-sequencing reads and quantification of gene expression as transcripts per million (TPM) was conducted using QuanTIseq, which utilises Trimmomatic and Kallisto, respectively, for those two steps. The resulting gene expression matrix was then analysed using CIBERSORT, BLADE and NITUMID.

CIBERSORTx: Signature matrix LM22, containing expression data for 547 genes and 22 immune cell types, was applied as a reference for the deconvolution algorithm. B-mode batch correction was applied to account for technical differences between the microarray data-based signature matrix and input gene expression matrix originating from bulk RNA-seq reads, quantile normalisation was disabled as recommended for RNA-seq data, and 100 permutations were set for robust statistical analysis. For subjects with repeated RNA-sequencing experiments, first repeats were chosen arbitrarily for subsequent analysis. Estimated proportions of CD8+ T-cells were extracted from the output file containing estimate fractions for 22 immune cell types and grouped depending on tumour response.

The BLADE deconvolution tool was applied as per the author-provided guidance available at https://github.com/tgac-vumc/BLADE. Briefly, BLADE was installed using Conda, input variables created as per guidance and author-provided scripts modified to define input and output files. Deconvolution was performed four times, each using one of the four author-provided signature matrices, allowing for distinguishing between 4 and 15 immune cell types. Default hyperparameters were applied in the phase of Empirical Bayes. A default number of repeats for evaluating each parameter configuration in Empirical Bayes phase (Nrep) and a default number of repeated optimizations for the final parameter set (Nrepfinal) were applied. Number of jobs executed in parallel (Njob) was increased to 50.

NITUMID was utilised for analysis in R 4.2.1 to as per the author-provided guidance available at https://github.com/tdw1221/NITUMID. An author-provided signature matrix was applied, allowing for distinguishing between the total of 11 cell types.

## Gene set enrichment analysis

The gene set enrichment analysis (GSEA) was performed by applying the fGSEA R package. Row read counts were normalized and all genes ranked according to Wald test *p-value* by using the DESeq2 R package. For multiple correction, the false discovery rate (FDR) approach of Benjamini–Hochberg adjusted *P*-value was applied, and <0.05 was considered significant. The visualisation of the results was conducted with the help of the ggplot2 R package.

## 16S RNA gene sequencing and gut microbial community analysis

Stool samples for 16S RNA sequencing were collected prior to initiation of treatment using ISO13485:2016 accredited sample collection kits (Atlas Biomed, London, UK). 16S rRNA genes in 16S V3-V4 regions were amplified with specific and barcodes. The 16S primer sequences are proprietary to Atlas Biomed and therefore confidential. All PCR mixtures contained 15 μL of Phusion® High-Fidelity PCR Master Mix (New England Biolabs, Ipswich, MA, USA), 0.2 μM of each primer and

10 ng target DNA., After PCR composed of 30 cycles at 98 °C (10 s), 50 °C (30 s) and 72 °C (30 s) and a final 5 min extension at 72 °C the PCR products were analysed on 2% agarose gel. Finally, PCR products were purified with n Qiagen Gel Extraction Kit (Qiagen, Hilden, Germany) following manufacturer's recommendations. NEBNext® UltraTM IIDNA Library Prep Kit (New England Biolabs, Ipswich, MA, US Cat No. E7645) was used for generating sequencing libraries and library quality was evaluated on the Qubit@ 2.0 Fluorometer (ThermoFisher Scientific, Waltham, MA, USA) and Agilent Bioanalyzer 2100 system (Agilent Technologies, Santa Clara, CA, US). Finally, the library was sequenced on an Illumina platform (NanoSeq Illumina, San Diego, CA, US) and 100 bp single-end reads were generated.

## Gut microbial community analysis

Fastp (-q 19 -u 15 -n 25 -l 60 --min_trim_length 10) software was used to do quality control of single-end raw data and to generate high-quality sequencing reads. Vsearch software was used to blast clean reads to the silva (v138) database to detect the chimera and remove them, so as to obtain the final effective data, namely effective tags. For the effective tags, the deblur module in QIIME2 software was used to do denoise, and the sequences with less than 5 abundance were filtered out to obtain the final ASVs (Amplicon Sequence Variables) and feature tables. Then, the Classify-sklearn module in QIIME2 software was used to compare ASVs with the database and to obtain the taxonomical classification of each ASV. Specificity was 97% to the genus level and only 80% to species level. The Boruta algorithm was employed to discern genera that exhibit significant associations with either the R or NR subgroups. A $p$-value < 0.05 was carried out using the Mann–Whitney test, illustrated using box plots of absolute microbial abundance. A microbiota rheostat was computed from the logartithm (base 10) of the ratio of the sum of the absolute abundances corresponding to significant genera in the R subgroup ($G_R$) divided by the sum computed for NR-subgroup ($G_{NR}$), ie.

$$Log_{10} \frac{\sum G_R}{\sum G_{NR}} \qquad (2)$$

Hereafter, the rheostat is referred to as $\log(G_R/G_{NR})$ in the manuscript. Spearman's rank correlation coefficient was used to determine the association between $\log(G_R/G_{NR})$ and tumour-intrinsic features, taking a $p$-value of <0.05 as significant.

## Microbiota diversity

Gut microbial alpha diversity was computed in QIIME2 using 8 methods; Among them, 1. chao1 (Chao1 index), 2. dominance (Berger-Parker Dominance index), 3. observed_otus (Number of distinct features) a richness index, 4. goods_ coverage (Good's coverage of counts), 5. pielou_evenness (Pielou's evenness) a measure of coverage and relative evenness of species. 6. shannon (Shannon's index) and 7. simpson (Simpson's index) an indicator of microbiota diversity, and 8. faith_pd (Faith's phylogenetic diversity), also anther diversity index that incorporates phylogenetic difference between species.

## Log($G_R$/$G_{NR}$) receiver operating characteristic and K-fold cross-validation

Area under the receiver operating curve (ROC-AUC) was created using the package 'pROC', version 1.18.0. K-fold cross-validation was performed with $k = 3$ equally sized folds, sampling without replacement using the 'sample()' function (the sample fraction 0.8 for the training, 0.2 for the testing), iterated until all folds were tested. The process was repeated 3 times. Mean accuracy mean score AUC were calculated. The summary metrics accuracy, AUC and the AUC 95% confidence interval were calculated.

## Gut microbiota - Linear discriminant analysis effect size

To identify significantly different bacteria between the two responses to immunotherapy at the genus level, LEfSe[20] (version 1.0.8) was performed using the default setting. Significance was set at $p$-value < 0.05 and LDA score cut-off point of 2. A cladogram representative of the structure of the R and NR microbiota was generated.

## Gut microbiota - functional gene prediction

PICRUSt2 (Phylogenetic Investigation of Communities by Reconstruction of Unobserved States, v2.0) was used to predict the functional gene content of each ASV sequence. The predicted functions were calculated in TIGRFAM, Enzyme Commission number (EC), metabolic pathways (MetaCyc) and KEGG pathways.

## Immunohistochemistry

P16ink4a, *BAP1*, BRCA1, and PD-L1 (CD274) immunohistochemistry were conducted. As controls, normal tonsil tissue was used for p16ink4a, normal breast tissue for BRCA1, a positive cell line for *BAP1*, and tonsil and negative or positive cell lines for PD-L1 (clone 22C3 pharmDx, Agilent, Santa Clara, CA, USA). Stainings were evaluated independently by two experts. Sections exhibiting medium to strong antibody expression (≥10% of cells) were classified as positive for *BAP1*, p16ink4a, and BRCA1: for PD-L1 staining of >1% tumour proportion were scored positive.

## Multiplexed immunofluorescence

**Staining protocol.** FFPE sections from baseline (pre-treatment) biopsies were deparaffinised and rehydrated using standard procedures. For heat-induced epitope retrieval sections were microwaved in 10 mM Tris/1 mM EDTA (pH 9.0) for five minutes, followed by 15 min at 30% power. Epitope-retrieved sections were mounted onto Sequenza hydrophobic clips (ThermoFisher Scientific, Waltham, MA, USA) and stained using an Opal 6-Plex Manual Detection Kit (Akoya Biosciences, Marlborough, MA, USA) according to manufacturer's instructions.

Briefly, sections were blocked with 1× Antibody Diluent/Block for 10 min and stained with primary antibodies (diluted in PBS) for 30 min at room temperature, followed by secondary incubation with 1× Opal Anti-Ms+Rb HRP polymer for 30 min. Primary antibodies and opal fluorophores are in Table 1.

Fluorescence signals were developed by 10-min incubation with Opal fluorophore (480, 520, 570, 620, or 690) diluted at 1:100 in 1× Plus Amplification Diluent. Multiplexing was achieved by iterating this

## Table 1 | Primary antibodies used in the study

| Marker | Antibody clone | Antibody dilution | Supplier | Paired Fluorophore | Staining panel |
|---|---|---|---|---|---|
| CD4 | 4B12 | 1:50 | Dako | Opal 520 | 1 |
| CD8 | C8/144B | 1:200 | Dako | Opal 480 | 1 |
| CD45RO | UCHL1 | 1:600 | Dako | Opal 570 | 1 |
| CD45RA | 4KB5 | 1:1000 | Santa Cruz Biotechnology | Opal 780 | 1 |
| CD68 | KP-1 | 1:500 | Dako | Opal 620 | 2 |
| VISTA | D1L2G | 1:200 | Cell Signaling Technology | Opal 780 | 2 |

process for each primary antibody/Opal fluorophore pair. For the final (sixth) round of staining, Opal TSA-DIG was used instead of fluorophores, followed by Opal 780 fluorophore incubation (1:50 dilution in 1× Antibody Diluent/Block) for 60 min. Slides were then counterstained with 4′6-diamidino-2-phenylindole (DAPI, 6 μM) for 5 min and mounted using ProLong™ Diamond mounting media (ThermoFisher Scientific, Waltham, MA, USA). Sections similarly treated with omission of fluorophore/DAPI incubation were used for autofluorescence compensation in downstream image processing.

### Digital image analysis
Whole slide scanning was performed using a Vectra Polaris™ (Akoya Biosciences, Marlborough, MA, USA) automated quantitative pathology imaging system (multispectral slide scan mode with 0.50 μm pixel resolution), according to manufacturer's instructions. Acquired whole scan image files were imported into inForm 2.6.0 image analysis software (Akoya Biosciences, Marlborough, MA, USA), and quantitative image analysis was performed with following steps:
(1) auto-fluorescence compensation;
(2) cell (nuclear) segmentation based on DAPI staining;
(3) single-cell phenotyping based on multiplex marker staining.

Cell segmentation algorithms were trained using over 15 independent mesothelioma tissues obtained prior to this study, and the trained algorithms were further fine-tuned with each image file used in this study. To calculate the percentage of the cells with each phenotype (e.g. CD45RO+CD8+ cytotoxic T cells, CD68+VISTA+ macrophages), automated cell phenotyping/counting using the fine-tuned algorithms was performed throughout the tissue in an unbiased manner, and single-cell data-outputs containing not only phenotypes but also positional and fluorescence intensity information were analysed using Python (version 3.6, package pandas 1.1.5).

The numbers and percentages of the cells with each phenotype were determined for each patient. Random forest-based feature selection analysis was run on the percentages of the cells to identify the cell phenotypes best correlated with tumour response (Boruta_py 0.3, 5000 maximum iteration, p value threshold 0.05). Features indicated by Boruta algorithm were further tested using the Wilcoxon rank sum test, unpaired, 2 sided with p value threshold of 0.05 for significance (Graphpad prism 9.4.1). CD4 and CD8 markers were selected as important feature for further testing regardless of selection done by Boruta.

### Machine learning
Random forest classification was used to select relevant variables. The importance of each variable was compared with the maximum importance value of all random (shadow) features using a permutation test. Analysis was repeated 10 times using 5000 iterations each. The R statistical software version 4.1 and the 'ranger' package were used for random forest training and variable importance elimination.

### Biomarker analysis
Continuous variables per response category were compared using non-parametric unpaired testing (Wilcoxon), two-sided with a significance threshold $p$-value of 0.05. For categorical variables, Fisher exact test was used significance threshold of p = 0.05. Spearman's rank correlation was used for analysis of continuous variable association with a p-value significance threshold of 0.05. Survival analysis used Kaplan–Meier survival using logrank estimation of median time to event parameters (overall and progression-free survival). $P$-values were estimated using Mantel–Cox testing with a $p$-value of <0.05 as the threshold for significance. Cox-proportional regression analysis was used to calculate hazard ratios. Post-hoc Analyses employed R version 4.3.2 and Prism 9.5.1 (Graphpad, San Diego, CA, USA). Illustrations were created with Biorender.com.

### Reporting summary
Further information on research design is available in the Nature Portfolio Reporting Summary linked to this article.

## Data availability
Patient-related data related to this clinical trial shall remain confidential to the sponsor organisation (The University of Leicester) and will not be disclosed except where disclosure might be required in accordance with pharmacovigilance duties of the parties involved. Individual participant data can be made available, after deidentification to investigators who provide written request in accordance with General Data Protection Regulation and following authorisation from the sponsor organisation, starting immediately and ending 3 years after publication. Requests for data and materials will be reviewed by the sponsor and any implications regarding intellectual property or confidentiality considered The WES and RNA-sequencing raw data is available in SRA Run Selector. The data can be publicly accessed upon publication via https://www.ncbi.nlm.nih.gov/bioproject/PRJNA916814, which is hosted by the National Centre for Biotechnology Information, under accession number PRJNA916814. All of the other data supporting the findings of this study are available within the article and its supplementary information files and from the corresponding author upon reasonable request. Source data are provided with this paper.

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

## Acknowledgements

The research was carried out at the National Institute for Health and Care Research (NIHR) Leicester Biomedical Research Centre (BRC) For the purpose of open access, the author has applied a Creative Commons Attribution license (CC BY) to any Author Accepted Manuscript version arising from this submission. The MiST study is sponsored by the University of Leicester (Leicester, UK) and funded by the Asthma + Lung UK and British Lung Foundation Partnership and the Victor Dahdaleh Foundation (Toronto, ON, Canada; grant VPDCF17-17). The study drugs (Atezolizumab and Bevacizumab) were provided by Roche Oncology (Basel, Switzerland). Additional funding is provided by Cancer Research UK in conjunction with the UK Department of Health on Experimental Cancer Medicine Centre grant (C10604/A25151). We thank Adrian Nicholson, Judith Underwood, Azmina Sodha-Ramdeen (University Hospitals of Leicester NHS Trust, Leicester, UK), Dr Caroline Cowley (Cancer Research Centre, University of Leicester, Leicester, UK), and Sam Moody (The Northern Cancer Research Centre, Newcastle, UK) for their valued contribution to this study. Furthermore, we acknowledge the Advanced Imaging Facility at the University of Leicester (RRID:SCR_020267) for their support. We especially thank the patients and their families who were involved in this clinical trial, and Mesothelioma UK. Jake Spicer was funded by an MRC Doctoral Training Partnership iCASE studentship.

## Author contributions

D.A.F. conceived the study. D.A.F., A. Branson, A.K., C.P., S.B., H.W., M.B., P.W.J., C.J.R. contributed to the study design, A. Bzura, M.Z., Z.Z., E.Y.B., J.S., T.K., J.D., J.H., N.N., D.F., Q.S., E.J.H., D.L., M.J., G.G., C.B., A.K., A.C., K.H., C.P., F.D., H.Y., J.C.H. contributed to data analysis, interpretation, writing, and manuscript editing. D.A.F., A.G., A.N., L.N., M.S., L.D., B.M., A. Bajaj, K.K., J.L.L., contributed to data collection.

## Competing interests

D.A.F. reports grants from Aldeyra, Astex Therapeutics, Bayer, BMS and Boehringer Ingelheim, Owkin; non-financial support from BerGenBio, Clovis, Eli Lilly, MSD, Roche, and Tesaro GSK; personal fees from Aldeyra, Cambridge Clinical Laboratories, Ikena, Opna Bio, Owkin, RS Oncology, Roche, MSD, during the conduct of the study. All other authors declare no competing interests.

## Additional information

Min Zhang[1,2,20], Aleksandra Bzura[1,20], Essa Y. Baitei[1,3,20], Zisen Zhou [4,20], Jake B. Spicer [1,20], Charlotte Poile[1], Jan Rogel[1], Amy Branson[5], Amy King[6], Shaun Barber[5], Tamihiro Kamata [1], Joanna Dzialo [1], James Harber[7], Alastair Greystoke [8], Nada Nusrat[1], Daniel Faulkner [1], Qianqian Sun[2], Luke Nolan[9], Jens C. Hahne[1], Molly Scotland[6], Harriet Walter [1,6], Liz Darlison[10], Bruno Morgan[11], Amrita Bajaj[11], Cassandra Brookes[5], Edward J. Hollox [12], Dominika Lubawska[12], Maymun Jama[1], Gareth Griffiths[13], Apostolos Nakas[14], Kudzayi Kutywayo[14], Jin-Li Luo[15], Astero Klampatsa [16], Andrea Cooper [17], Koirobi Halder[17], Peter Wells-Jordan[1], Huiyu Zhou[4], Frank Dudbridge [18], Anne Thomas[1,6], Catherine Jane Richards[19], Catrin Pritchard[1], Hongji Yang [4], Michael Barer[17] & Dean A. Fennell [1,6] ✉

[1]National Institute for Health Research Biomedical Research Centre & Cancer Research UK Experimental Cancer Medicine Centre, University of Leicester, Leicester, UK. [2]Novogene Corporation, Beijing, China. [3]Center for Genomic Medicine, King Faisal Specialist Hospital & Research Centre, Riyadh, Saudi Arabia. [4]Department of Informatics, University of Leicester, Leicester, UK. [5]Leicester Clinical Trials Unit, University of Leicester, Leicester, UK. [6]Department of Oncology, University Hospitals of Leicester NHS Trust, Leicester, UK. [7]Harry Perkins Institute of Medical Research and The University of Western Australia Centre for Cancer Research, Perth, WA, Australia. [8]Northern Centre for Cancer Care, Newcastle, UK. [9]University Hospital Southampton NHS Foundation Trust, Southampton, UK. [10]Mesothelioma, Leicester, UK. [11]Department of Radiology, University Hospitals of Leicester NHS Trust, Leicester, UK. [12]Department of Genetics and Genome Biology, University of Leicester, Leicester, UK. [13]University of Southampton Clinical Trials Unit, Southampton, UK. [14]Department of Cardiothoracic Surgery, University Hospitals of Leicester NHS Trust, Leicester, UK. [15]Bioinformatics and Statistics Analysis Hub, University of Leicester, Leicester, UK. [16]Institute of Cancer Research, Sutton, UK. [17]Department of Respiratory Sciences, University of Leicester, Leicester, UK. [18]Department of Health Sciences, University of Leicester, Leicester, UK. [19]Department of Pathology, University Hospitals of Leicester NHS Trust, Leicester, UK. [20]These authors contributed equally: Min Zhang, Aleksandra Bzura, Essa Y. Baitei, Zisen Zhou, Jake B. Spicer. ✉e-mail: df132@leicester.ac.uk

