## [Peer Review File · Nature Communications]

A Gut Microbiota Rheostat Forecasts Response to PD-L1-VEGF Blockade in MesotheliomaREVIEWER COMMENTS

Reviewer #1 (expert in respiratory medicine and lung "omics"):

Zhang and authors have conducted a phase II clinical trial in patients with mesothelioma with the aim to investigate the factors of favourable response so Immune Checkpoint Inhibitors (ICB). MIST4 recruited patients with relapsed (progressed after the initial treatment) mesothelioma (24 out of 26 had pleural mesothelioma). This is a small (n=26) but well-designed study. The authors identified aneuploidy, homologous recombination, epithelial to mesenchymal transition, and CD68+ macrophages as factors contributing to ICB resistance. Moreover, they identified gut bacterial patterns that positively correlated with radiological response to treatment and CD8+ T cell infiltration.

ICB has shown great promise in mesothelioma which is an incurable malignancy. The current NICE guidelines suggest the combination of chemotherapy and ICB as first line treatment. However, the mechanisms which determine the response to ICB remain unknown. It is an unmet clinical call to understand the factors which contribute to ICB resistance to better stratify the patients. The present study is significant and original.

I think it would be interesting to see in figure 1 and in the analysis some comparisons for the patients who had a had a clinically significant response in tumour size as described by the modified RECIST criteria. This might be tricky because of the small number of patients though. I would suggest to make some comparisons between patients who had a clinically significant response. Would the identified ICB resistance factors remain the same? For example in Figure 2A there appear to be seven patients who showed a response below the partial response. Three of these patients showed a minimal response.

In Figure 2C UPD there appears to be one patient in the NR group with the highest value which is probably driving the difference between the groups.

How was PD-L1 expression was determined in Figure 3B?

For Figure 3G how EMT low and high were determined? This is a very interesting results which I would suggest elucidating more.

In Figure 4B what is the Y axis? Does this refer to the reads?

The association of ICB resistance to the gut microbiology is very interesting. The 16S rRNA sequencing method is robust and could identify the total microbiome. I was wondering whether the authors used any negative controls to identify the kitome (the contamination of the samples by the DNA extraction kit). It is not very clear in the methods how you compensated for the false positives.

In conclusion this is an interesting study to understand the factors which contribute to ICB resistance.

Reviewer #2 (expert in cancer genomics):

This manuscript describes the utilization of the gut microbiota profile to predict the effects of PD-L1-VEGF blockade in malignant mesothelioma. This study was conducted in association with the clinical trial named MiST4, in which combinatory use of atezolizumab and bevacizumab (AtzBev) is evaluated for the treatment of malignant mesothelioma. In MiST4, the primary endpoint with 50% 12-week disease control was achieved with the tolerable adverse effects. As for the biopsy samples isolated from the 26 cases of the enrolled patients, the authors conducted a series of omics analyses. Whole exome sequencing, RNA sequencing and immunochemistry analyses collectively identified that aneuploidy, uniparental disomy (UPD), homologous recombination (HRD), epithelial mesenchymal transition (EMT) should be the indicators of resistance for the treatment, as the intrinsic factors of cancers. Infiltration of CD68-positive macrophages is also identified as a contributing factor. Even more importantly, the authors also examined the microbiomes of the patients and found that the profile of gut-resident microbial genera is positively correlated with radiological response to the treatment, as the extrinsic factor. Taken these results together, the authors concluded that these intrinsic and extrinsic factors may confer clinical sensitivity to AtzBev. Especially, the gut microbiota represents a potentially "modifiable" factor. Overall, I consider this manuscript should convey substantial novel and useful insights. Particularly, while the first part on

the intrinsic tumor factors appeared from a rather usual approach, the latter part appeared to be more intriguing. The followings are the issues which I believe would be helpful for further enriching the contents of the paper, and, especially, for easier reading for the readers.

Major points:

1. The format of the manuscript is not reader-friendly. Overall, at least to me, it would be easier if the sections were clearly separate into the introduction, the results, discussion and the methods sections. Also, it is not always easy to follow which of the presented results correspond to which of the descriptions in the text. Putting subheadings may be useful in the text. Supplementary files are rich, but too lengthy, when the non-expert readers try to read through when they would like to collect detailed information.

2. The biological rationale is limited to consolidate the claimed factors can collectively explain the diverse drug responses in vivo. For example, to evaluate the tumor microenvironments more directly, the spatial transcriptome analyses could be conducted, even starting from biopsy samples.

3. It should be possible and necessary to further resolve to what extent each of the intrinsic and extrinsic factors contribute the classification, if this is based on the random forest.

4. In fact, in the text, the descriptions related to the clinical trials are rich. However, the descriptions on the experimental parts are not always sufficiently detailed. Particularly, I'm concerned about the quality of the generated datasets. For example, the quality of the RNA samples should be the key for several conclusions. When the RNAs isolated from FFPE samples are used, the quality of the material and the precise representation of the expression profiles as well as the uniform distribution of the sequence reads on the full-length mRNA should be more carefully evaluated. The same is true for the other omics analyses. For the whole exome sequencing, the sequencing depth and the coverage to the variant allele frequency should be described. The description on the microbiome analysis is perhaps the poorest part. For this part, please describe the DNA extraction method, primer information used for 16S rRNA amplification. Statistics of the obtained sequence data, which should be unified between samples for the fair statistical analyses, should be presented somewhere. All the parameters of the tools used for information analysis, such as fastp, vsearch, and QIIME2, should be also specified.

Minor points:

5. To claim the "absence" of the several factors, such as the gene fusions, the whole exome sequencing and the RNA sequencing (of a possibly limited quality) should not be enough. At least for some cases, whole genome sequencing should be conducted, desirably by long read sequencing and RNA sequencing of high quality, although operation materials or autopsy samples may be needed for this purpose.

6. To characterize the immune cell profiles, the evidence is too shallow. Further detailed data should be presented in addition to the immunohistochemistry data.

7. Fig. 4A should also include the results of the common alpha diversity indices such as Shannon's index.

8. Line 223; the analysis of bacterial composition should be described as QIIME2, rather than PICRUST.

9. Lines 222-229, please explain in more details how the Fig. 4B was drawn from the multiple methods such as LefSe, random forest, and extreme gradient boosting.

10. It is not clear what "abundance" indicates on the vertical axis in Fig. 4B.

11. It is not clear how the results of Fig. 4C should be interpreted.

12. Generally, please improve the manner of the data presentation. For example:

- Please unify the notation method of the figures. There is no inner plot of box plot in Fig. 4A, but

it is shown in Fig. 4B.

-Some figures cannot be read due to low resolution (e.g. the legend of Fig.1A).

-Data itself seems not present sometimes (e.g. Lines 155-157 state that NR and R are free from bias due to background factors such as age and gender, but there is no data to support this notion).

• Legends of the figures seem sometimes missing. (e.g. Sup. Fig 9)

• The figure legend, vertical axis, and horizontal axis legend are occasionally insufficient (e.g.: Fig. 2D)

13. A new index called $\text{Log}(\text{GR}/\text{GNR})$ is interesting. In fact, I'm impressed to see that the difference between NR and R could be determined with a higher accuracy than the cancer genome mutation information in this case. Because of this importance, I consider it better to explain in more details how useful the constructed metric is in a general context. Specifically, I think Sup. Fig.9C should be brought to main Fig. 4F. In addition, it is generally necessary to prepare a validation cohort for model accuracy verification using AUC ROC, to render this results more convincing. If a validation cohort cannot be prepared, at least cross validation analysis should be employed to demonstrate that this scheme should be very powerful.

14. Is it possible to discuss the mechanism for therapeutic effects based on the functional changes in the bacterial flora shown in Fig. 5B?

15. If the authors discuss that microbiome is "modifiable", an actually plausible plan should be discussed.

Reviewer #3 (expert in biostatistics):

This reviewer thanks the authors for a very thorough description of the results assessed from the MIST4 study.

The Main section (results) is rather lengthy and difficult to digest at times. Context to some of the statements/paragraphs might ease in understanding. Additionally, it would make the paper more reproducible if the authors provided the type of significance test used every time a p-value was mentioned. The authors should also be conscious of making redundant statements in the main section (since it is lengthy). For instance, in line 124, they restate the same thing stated in the previous sentence. It does not need to be stated twice.

Lines 146 - 157 describe an analysis that was done on a subset of the 26 patients. However, this reviewer does not understand why the numbers mentioned in those lines is different than 26. Explanation for those differences should be included somewhere.

Lines 239 - 244 mention the results of a linear discriminant analysis. However, this method is not mentioned as being used in the methods section. A discussion of this method should be added to the methods section.

For the methods section, authors should also be cognizant of redundant paragraphs. The section of GSEA, for instance, is mentioned twice (and is not identical). The basic stats methods used are as well (lines 429-433). Lines 543-550 are also repeated (although I think they make more sense in the ML section than in the first section).

Both the logistic regression section (lines 793-800) and the survival analysis section (lines (805-808) need expansion. For the logistic regression, it is unclear what the outcome/response variable used it. It is also unclear what the predictor/exploratory variables are. These should both be mentioned. It is also not clear from reading the Main section where logistic regression was actually used. If this method was not utilized, it should be removed from the methods section. If it was used, the results should be clarified to indicate they were from logistic regression and potentially include the implications of the models built (and possibly show the beta coefficients and/or odds ratios and confidence intervals from those models if relevant). For the survival analysis, the results

sections mention hazard ratios and Mantel-cox p-values. Those methods do not appear to be included in the methods section.

Also, many different versions of R are used throughout the methods section, as well as multiple different software packages (STATA, Prism/GraphPad, R, and python). This comes across as multiple different versions of the data being run at different times and different tests being done by different people which might lead to errors along the way. The analyses should be re-checked to ensure the appropriate data was used in each instance and carried forward to the next step as appropriate.

Reviewer #4 (expert in thoracic oncology):

In this manuscript, Dr. Fennell's team reports findings from correlative analyses of specimens from 26 pts participating in the phase II MIST4 study of atezolizumab + bevacizumab in pleural (n=24) and peritoneal (n=2) mesothelioma. The authors conducted multi-omic analyses (exome, transcriptome, gut microbiome, multiplex special immunofluorescence). Activity of the combination was modest with an ORR of 3.8% but DCR was 27% (n=7/26) at 24 weeks and 50% at 12 weeks. AEs led to treatment discontinuation in 15% of pts. This is a well-done study with robust correlatives from a group that is leading the field in mesothelioma targeted therapies.

Comments

- The authors mention a grade 5 AE of dyspnea. Please clarify whether this was due to pneumonitis or underlying disease.
- The authors should consider mentioning the response assessment criteria earlier on in the manuscript (modified RECIST), as well as stating the interval for imaging.
- The comparison of "responders" vs non-responders is somewhat limited by the absence of robust and durable responses to treatment.
- The authors should state the biopsy requirements earlier on in the manuscript to help readers contextualize the data
- The overall rate of BAP1 deficiency (by protein expression) seems lower than in an all-comer population of patients with mesothelioma.
- Due to the overall rarity of LATS1 mutations in mesothelioma and the small size of the cohort, it is somewhat challenging to state that LATS1 is enriched in non-responders
- The finding that HR deficiency correlated with non-response contrast findings from other groups (Forde Nature Medicine pre0505, Minchom et al PMID 32169873). What is the hypothesis behind this finding?
- The independent mesothelioma cohort (MEDUSA) is introduced abruptly. The authors should clarify why this dataset was only used for some of the analyses.
- The authors should include mention of the recently published UNITO-001 study of immunotherapy + PARPi in the discussion (Passiglia et al Clinical Cancer Research)
- 9p21 is mentioned as encompassing CDKN2A without any mention of CDKN2B or MTAP. Recent work suggests that the immunotherapy resistance may be also be due to MTAP in other cancers (PMID 37774699, Gjuka et al)
- Was there any difference in gut microbiota for the 2 peritoneal cases or for patients with abdominal involvement of their cancer versus those without? Was there any difference in patients who were heavily pretreated with chemotherapy vs those who were not? This may inform timing of this regimen.
- The spider plot in figure 1 is blurry, as is supplementary figure 2 and supplementary table 8b
- In the table, why are some patients listed as "not applicable" for p16 and BAP1 expression. Was there insufficient tissue in these cases?
- For Figure 2B, the number of patients in each group should be listed under the KM curve

RESPONSE TO REVIEWER COMMENTS

Reviewer #1 (expert in respiratory medicine and lung "omics"):

Reviewer 1's comment

1. I think it would be interesting to see in figure 1 and in the analysis some comparisons for the patients who had a had a clinically significant response in tumour size as described by the modified RECIST criteria. This might be tricky because of the small number of patients though. I would suggest to make some comparisons between patients who had a clinically significant response.

Response to reviewer's comment

The partial response rate measured by modified RECIST was observed in *only* 1/26 patients ie. 3.8% 95% CI 0.1-19.6) as shown in figure 1A. As such, and as reviewer 1 correctly suggests, there is insufficient power to enable a statistical comparison between this one significant responder and for example the 8/26 (3.8%) of patients who had progressive disease as their best response.

For our analysis we encompassed all assessable patients due to the small size of the phase 2 trial, dichotomising on whether shrinkage (the R group) or progression (NR) was observed. We show in figure 2B that this dichotomisation was associated with longer progression free survival and is therefore clinically relevant.

To address this reviewer's comment in the manuscript, a sentence has been added to say that due to the low mRECIST partial response rate, an analysis of extreme phenotypes compared with the mRECIST disease progression subgroup was not statistically feasible (line 120).

Reviewer 1's comment

2. Would the identified ICB resistance factors remain the same? For example in Figure 2A there appear to be seven patients who showed a response below the partial response. Three of these patients showed a minimal response.

Response to reviewer's comment

To confirm, there is only one patient who reached the mRECIST criterion of partial response. However approximately half of patients had stable disease or a reduction in tumour size not reaching the threshold for partial response (ie. Below or equal to 0% change in tumour volume). We have defined this group the “reduction” or R group and growth as the best outcome in the no-reduction (NR) group. To highlight the relative relationships between these groups of patients at the individual level, we colour coded NR as red and R as green. We find that these groups are phenotypically homogeneous and only present significant differences in the manuscript. The sample size is small and potentially sensitive to removal of data. As suggested by the reviewer an amendment has been added for clarification at line 162 in the manuscript.

Reviewer 1’s comment

3. In Figure 2C UPD there appears to be one patient in the NR group with the highest value which is probably driving the difference between the groups.

Response to reviewer’s comment

Yes this correct, however as shown in the box plot there is also wider spread in the lower UPD values for the NR group in comparison to the R group, where these values are consistently low. We accept that there are limitations due to the small sample size however we have presented this result as the p value achieved the significance threshold of 0.05. A sentence addressing this observation has been made in the results section (line 172).

Reviewer 1’s comment

4. How was PD-L1 expression was determined in Figure 3B?

Response to reviewer’s comment

PDL-1 expression was determined using the PDL1 22C3 tumour proportion score. Although this is detailed in the methods, a sentence is added to highlight this in the main text (line 210).

Reviewer 1’s comment

5. For Figure 3G how EMT low and high were determined? This is a very interesting results which I would suggest elucidating more.

Response to reviewer's comment

We calculated with enrichment score for EMT corresponding to the hallmark geneset enrichment analysis (GSEA) signature; this transcriptional signature was found to be significantly enriched in the NR group. A sentence addressing reviewer 1's question has been added to the main text (line 238).

Reviewer 1's comment

6. In Figure 4B what is the Y axis? Does this refer to the reads?

Response to reviewer's comment

Correct. The figure 4B has been modified to state "absolute abundance of microbiota" which correlates with reads - this label has now been added as a generalised y axis reference).

Reviewer 1's comment

7. The association of ICB resistance to the gut microbiology is very interesting. The 16S rRNA sequencing method is robust and could identify the total microbiome. I was wondering whether the authors used any negative controls to identify the kitome (the contamination of the samples by the DNA extraction kit). It is not very clear in the methods how you compensated for the false positives.

Response to reviewer's comment

The methods describing the microbiome analysis have been re-written for clarity. We used a research service (Atlas London, UK) as the service provider. Negative controls were not employed in the analysis, however the microbiome analysis was conducted to ISO13485:2016 accreditation for medical device quality management systems using DNA microarray technology (illumina) in a certified EU laboratory.

Reviewer #2 (expert in cancer genomics):

Reviewer 2's comment

1. Major points:

The format of the manuscript is not reader-friendly. Overall, at least to me, it would be easier if the sections were clearly separate into the introduction, the results, discussion and the methods sections.

Response to reviewer's comment

To address this important point, subheadings have now been added accordingly as suggested . The methods subheading is now highlighted in Arial Bold 24

Reviewer 2's comment

2. Also, it is not always easy to follow which of the presented results correspond to which of the descriptions in the text. Putting subheadings may be useful in the text. Supplementary files are rich, but too lengthy, when the non-expert readers try to read through when they would like to collect detailed information.

Response to reviewer's comment

Agreed. Additional subheadings have been added to the main text which corresponding to the results that are shown in each figure. As such there are now 7 subheadings added to the results section

Reviewer 2's comment

3. The biological rationale is limited to consolidate the claimed factors can collectively explain the diverse drug responses in vivo. For example, to evaluate the tumor microenvironments more directly, the spatial transcriptome analyses could be conducted, even starting from biopsy samples.

Response to reviewer's comment

The translational research conducted in this study utilised the very small and limited amount of tissue available from (mostly) thoracoscopic biopsies. As such, much of the tissue has been consumed during these studies and there

therefore spatial transcriptomic analysis is not possible. It should also be noted that this study was an investigator-initiated clinical trial with a limited budget that could not accommodate the high costs of spatial transcriptomic analysis. However this is a potentially powerful technology that we are definitely keen to explore this in future studies.

Reviewer 2's comment

4. It should be possible and necessary to further resolve to what extent each of the intrinsic and extrinsic factors contribute the classification, if this is based on the random forest.

Response to reviewer's comment

We have compared the relative predictive performance of intrinsic versus extrinsic factors figure 4F using ROC analysis. The Area under curve for Log G_R/G_{NR} is largest compared with either UPD or HRD, suggesting that this factor makes a greater contribution to the classification. However this does not exclude the possibility that they are co-variate ; a sentence has been therefore added to the discussion to address this point at line 349.

Reviewer 2's comment

5. In fact, in the text, the descriptions related to the clinical trials are rich. However, the descriptions on the experimental parts are not always sufficiently detailed. Particularly, I'm concerned about the quality of the generated datasets. For example, the quality of the RNA samples should be the key for several conclusions. When the RNAs isolated from FFPE samples are used, the quality of the material and the precise representation of the expression profiles as well as the uniform distribution of the sequence reads on the full-length mRNA should be more careful evaluated. The same is true for the other omics analyses.

Response to reviewer's comment

Thank you. In accordance with this reviewer's recommendation we have extensively revised the text with addition of a supplementary quality controls table including both whole exome and RNA sequencing alignment metrics (referenced as the supplementary data table S1 at line 157).

Reviewer 2's comment

6. For the whole exome sequencing, the sequencing depth and the coverage to the variant allele frequency should be described.

Response to reviewer's comment

This information is now provided in the supplementary data table S4 which is highlighted in the results at line 157

Reviewer 2's comment

7. The description on the microbiome analysis is perhaps the poorest part. For this part, please describe the DNA extraction method, primer information used for 16S rRNA amplification. Statistics of the obtained sequence data, which should be unified between samples for the fair statistical analyses, should be presented somewhere.

Response to reviewer's comment

16S RNA extraction and sequencing is revised in the methods at line 696. Sequencing was conducted as a research service by Atlas Biomed (London, UK) in accordance with ISO13485:2016. Primer sequence information was proprietary and confidential. A statement regarding this is included in the methods (line 711)

Reviewer 2's comment

8. All the parameters of the tools used for information analysis, such as fastp, vsearch, and QIIME2, should be also specified.

Response to reviewer's comment

The method has been revised accordingly to include the tool parameters for these applications (from line 735)

Reviewer 2's comment

9. Minor points:

To claim the "absence" of the several factors, such as the gene fusions, the whole exome sequencing and the RNA sequencing (of a possibly limited quality) should not

be enough. At least for some cases, whole genome sequencing should be conducted, desirably by long read sequencing and RNA sequencing of high quality, although operation materials or autopsy samples may be needed for this purpose.

Response to reviewer's comment

Thank you for this comment. Unfortunately there was insufficient tissue available from the small diagnostic biopsies to allow for whole genome sequencing. A statement stating the limitation of the bulk RNA sequencing method used and lack of whole genome data availability on this cohort is given in the discussion (line 193).

Reviewer 2's comment

10. To characterize the immune cell profiles, the evidence is too shallow. Further detailed data should be presented in addition to the immunohistochemistry data.

Response to reviewer's comment

Quantitative immunophenotyping in MIST4 was limited to a 6 colour, custom multiplex immunofluorescence-based analysis. We utilised machine learning based filtering of immune antigen expression and present *only* the statistically significant results. These findings were orthogonally validated using bulk RNA sequencing based deconvolution, employing CIBERSORT, BLADE and NITUMID. Although multiplex immunofluorescence has been rapidly advancing since MIST4, we provide cross-validated results implicating CD8 and CD68 positive tumour infiltrating leucocytes as response associated-intrinsic factors. A statement in the discussion has been added to discuss the limitations of the multiplex panel used in this study (337).

Reviewer 2's comment

11. Fig. 4A should also include the results of the common alpha diversity indices such as Shannon's index.

Response to reviewer's comment

Thank for this query. We have thoroughly reviewed the microbiota alpha diversity measures and in addressed this query have now added new results computed using 8 orthogonal methods (line 245) - the Shannon index, Simpson's index Chao1 index, Berger-Parker dominance index, Richness index (Observed-otus), Good's coverage of counts, Pielou's evenness, and Faith's phylogenetic diversity. These were computed in QIIME2. We originally computed Hill's index following a dimension reduction pre-processing step using random forests. Although, dimension reduction has been used for this kind of analysis in the literature, this result has been superseded in this revision in favour of 8 cross validated methods which all agree and show no significant difference in alpha diversity between R and NR subgroups. A new supplementary figure (box plots) has been added as supplementary figure , and the results text amended at line 245 to reflect these changes. Diversity score analysis is hindered by high dimensionality in a small sample, which likely underpins this result and this is therefore highlighted at line 249.

Reviewer 2's comment

12. Line 223; the analysis of bacterial composition should be described as QIIME2, rather than PICRUST.

Response to reviewer's comment

Correct. An amendment to the text at line 251 has been made and this is highlighted in the methods (line 732)

Reviewer 2's comment

13. Lines 222-229, please explain in more details how the Fig. 4B was drawn from the multiple methods such as LefSe, random forest, and extreme gradient boosting.

Response to reviewer's comment

Figure 4B shows the difference in absolute microbiota abundance for each of the significantly enriched genera in R versus NR subgroups respectively. The results shown have been verified as significant using non-parametric Mann

Whitney test with p values < 0.05. These genera were discovered using orthogonal methods ie. random forest analysis and LEfSe.

In the text, only the random forest p values are shown (line 253). A table of p values across these methods corresponding to the overlapping genera are shown in a new supplementary table 8C.

Reviewer 2's comment

14. It is not clear what "abundance" indicates on the vertical axis in Fig. 4B.

Response to reviewer's comment

This has been addressed above in response to reviewer 1, with an amendment to include the term "Absolute microbiota abundance" on a generalised y axis. The term is also clarified in the methods at line

Reviewer 2's comment

15. It is not clear how the results of Fig. 4C should be interpreted.

Response to reviewer's comment

Figure 4C is a standard cladogram, showing the relative phylogenetic distance of the significant genera associated with treatment response that were identified in MIST4. To ensure clarity, a statement regarding this is highlighted in the figure legend corresponding to 4C (line 257).

Reviewer 2's comment

16- Please unify the notation method of the figures. There is no inner plot of box plot in Fig. 4A, but it is shown in Fig. 4B.

Response to reviewer's comment

4B and 4D have been amended to match all other figures 4A, 2C, 2F,3E,3F

Reviewer 2's comment

17. Some figures cannot be read due to low resolution (e.g. the legend of Fig. 1A).

Response to reviewer's comment

Figures 1A, 1B have been replaced with higher resolution panels.

Reviewer 2's comment

18. Data itself seems not present sometimes (e.g. Lines 155-157 state that NR and R are free from bias due to background factors such as age and gender, but there is no data to support this notion).

Response to reviewer's comment

As part of the clinical statistical analysis plan, the MIST4 trial statisticians, computed the association with response for these clinical features. As these results are not significant, the results are not explicitly shown in the manuscript. Due to a lack of statistical significance related to these factors, and therefore relevance, this statement has been removed for clarity.

Reviewer 2's comment

19. legends of the figures seem sometimes missing. (e.g. Sup. Fig 9)

Response to reviewer's comment

This has been corrected – the supplementary figures have been correctly re-labelled

Reviewer 2's comment

20. The figure legend, vertical axis, and horizontal axis legend are occasionally insufficient (e.g.: Fig. 2D)

Response to reviewer's comment

Figure 2D has been modified to include labels for the X axis (MIST patient identifier, Y axis chromosomal locus), similarly for 2E labels have been added (X axis, patient identifier, y axis driver)

Reviewer 2's comment

21. A new index called $\text{Log}(\text{GR}/\text{GNR})$ is interesting. In fact, I'm impressed to see that

the difference between NR and R could be determined with a higher accuracy than the cancer genome mutation information in this case. Because of this importance, I consider it better to explain in more details how useful the constructed metric is in a general context. Specifically, I think Sup. Fig.9C should be brought to main Fig. 4F.

Response to reviewer's comment

Thank you. Figure 9C compares the predictive performance using receiver operating characteristic (ROC) analysis for intrinsic versus extrinsic features and this has been brought into the main figures replacing figure 4F. This figure also addresses an earlier comment by reviewer 1 relating to the relative predictive performance of extrinsic versus intrinsic factors (UPD and HRD).

Reviewer 2's comment

22. In addition, it is generally necessary to prepare a validation cohort for model accuracy verification using AUC ROC, to render this results more convincing. If a validation cohort cannot be prepared, at least cross validation analysis should be employed to demonstrate that this scheme should be very powerful.

Response to reviewer's comment

Agreed. The ROC analysis has been revised to now include a k-fold cross validation (added to the results at line 262 and methods at line 759). In the main results, the average AUC estimated from k-fold cross validation is revised (0.94) and a table comprising the log G_R/G_{NR} ROC AUC metrics are now included in the supplementary table – comprising the mean accuracy, AUC and associated 95% and 99% confidence limits

Reviewer 2's comment

23. Is it possible to discuss the mechanism for therapeutic effects based on the functional changes in the bacterial flora shown in Fig. 5B?

Response to reviewer's comment

At present it is not possible to assign a causal link between the biochemical pathway enrichment in R versus NR and response to immunotherapy. These

pathways are observational but may serve as hypothesis generating for future investigation. A statement reflecting this is made in the discussion (line 357)

Reviewer 2's comment

24. If the authors discuss that microbiome is “modifiable”, an actually plausible plan should be discussed.

Response to reviewer's comment

Spencer et al Science 2021, vol 374 1632 reported that human dietary modification with fibre (not probiotics) can modulate treatment response to ICB. This citation has been included and the statement around the term modifiable, amended to include dietary modification and this citation.

Reviewer #3 (expert in biostatistics):

Reviewer 3's comment

1. This reviewer thanks the authors for a very thorough description of the results assessed from the MIST4 study.

The Main section (results) is rather lengthy and difficult to digest at times. Context to some of the statements/paragraphs might ease in understanding.

Response to reviewer's comment

Subtitles have been added as suggested by reviewer 2 and addressed above

Reviewer 3's comment

2. Additionally, it would make the paper more reproducible if the authors provided the type of significance test used every time a p-value was mentioned. The authors should also be conscious of making redundant statements in the main section (since it is lengthy). For instance, in line 124, they restate the same thing stated in the previous sentence. It does not need to be stated twice.

Response to reviewer's comment

The significance test is now mentioned throughout in the results (and also described in the methods section. Repeated text has been thoroughly reviewed and amended accordingly including the section highlighted by reviewer 3

Reviewer 3's comment

3. Lines 146 - 157 describe an analysis that was done on a subset of the 26 patients. However, this reviewer does not understand why the numbers mentioned in those lines is different than 26. Explanation for those differences should be included somewhere.

Response to reviewer's comment

The clinical trial enrolled 26 patients however our translational research studies involved 20 patients (whole exome), 20 patients (RNA sequenced),

multiplex immunofluorescence in 26 patients and 16S RNA sequencing in 24 patients. This information is now included in the results at line.

Reviewer 3's comment

4.Lines 239 - 244 mention the results of a linear discriminant analysis. However, this method is not mentioned as being used in the methods section. A discussion of this method should be added to the methods section.

Response to reviewer's comment

This method is now added in the methods section at line 750

Reviewer 3's comment

5.For the methods section, authors should also be cognizant of redundant paragraphs. The section of GSEA, for instance, is mentioned twice (and is not identical). The basic stats methods used are as well (lines 429-433). Lines 543-550 are also repeated (although I think they make more sense in the ML section than in the first section).

Response to reviewer's comment

The Methods section has been thoroughly checked for redundancy and redundant text identified and removed, including those sections mentioned by reviewer 3.

Reviewer 3's comment

6. Both the logistic regression section (lines 793-800) and the survival analysis section (lines (805-808) need expansion. For the logistic regression, it is unclear what the outcome/response variable used it. It is also unclear what the predictor/exploratory variables are. These should both be mentioned. It is also not clear from reading the Main section where logistic regression was actually used. If this method was not utilized, it should be removed from the methods section. If it was used, the results should be clarified to indicate they were from logistic regression and

potentially include the implications of the models built (and possibly show the beta coefficients and/or odds ratios and confidence intervals from those models if relevant).

Response to reviewer's comment

Agreed. The classification of clinical outcome as either R or NR is a binary classification problem and although logistic regression can be used to explore the relationship between logGR/GNR and this binary response category, this is not essential for ROC analysis. For clarity therefore, we have re-analysed the ROC AUC using k-fold validation (line 759) as advised and the methods re-written, omitting any reference to logistic regression.

Reviewer 3's comment

7. For the survival analysis, the results sections mention hazard ratios and Mantel-cox p-values. Those methods do not appear to be included in the methods section.

Response to reviewer's comment

Agreed. Reference to the Mantel-Cox test has been now included in the methods section and is highlighted (line 868)

Reviewer 3's comment

8. Also, many different versions of R are used throughout the methods section, as well as multiple different software packages (STATA, Prism/GraphPad, R, and python). This comes across as multiple different versions of the data being run at different times and different tests being done by different people which might lead to errors along the way. The analyses should be re-checked to ensure the appropriate data was used in each instance and carried forward to the next step as appropriate.

Response to reviewer's comment

The versions of software were re-checked and the most up to date versions included in this revision. Analysis was re-checked with no evidence of any variations in the results. It should be noted that STATA was used by the clinical

trials statisticians as the platform for protocol-specified analyses. Post-hoc translational analyses used Graphpad prism as specified at line 869.

Reviewer #4 (expert in thoracic oncology):

Reviewer 4's comment

1. The authors mention a grade 5 AE of dyspnea. Please clarify whether this was due to pneumonitis or underlying disease.

Response to reviewer's comment

This was due to underlying disease. An amendment has been made to address this addition to the text and is highlighted (line 139)

Reviewer 4's comment

2. The authors should consider mentioning the response assessment criteria earlier on in the manuscript (modified RECIST), as well as stating the interval for imaging.

Response to reviewer's comment

Modified RECIST is now mentioned in the Efficacy section of the results with additional mention of this scanning interval. This is highlighted at line 116

Reviewer 4's comment

3. The comparison of "responders" vs non-responders is somewhat limited by the absence of robust and durable responses to treatment.

Response to reviewer's comment

To clarify, at line 162 we introduce the terms reduction (R) and non-reduction (NR) to classify the best response by modified RECIST, such that those patients who had any stabilisation or shrinkage of their cancer are in the R group and those with growth as their best response are in the NR group. This definition is quite distinct from mRECIST "response" which imply 30% reduction in tumour volume for responders and 20% growth for progression. This definition enables (in a small phase II, correlative studies) inclusion of the maximum number of patients in the cohort to be encompassed in the analysis to ensure sufficient power to identify statistically significant patterns in the data.

The definition of progression (growth beyond +20%) and response (tumour shrinkage below 30%) in modified RECIST is somewhat arbitrary. Biologically speaking, any reduction signifies some degree of sensitivity phenotype, whereas growth reflects an ICB-refractory phenotype. A statement to clarify the rationale for this cohort dechotomisation is made in the results section at line (line 162).

Reviewer 4's comment

4.The authors should state the biopsy requirements earlier on in the manuscript to help readers contextualize the data

Response to reviewer's comment

A sentence is now included in the results after describing the exome and transcriptome sequenced cohorts (at line 155).

5.The overall rate of BAP1 deficiency (by protein expression) seems lower than in an all-comer population of patients with mesothelioma.

Response to reviewer's comment

Thank you for highlighting this – the exome sequencing revealed an equal frequency of somatic alterations involving BAP1, and including UPD in NR (63%) versus R (6/10 patients 60%), p value 1 by Fisher exact test. This lack of a statistically significant difference is added at line 174.

Reviewer 4's comment

6.Due to the overall rarity of LATS1 mutations in mesothelioma and the small size of the cohort, it is somewhat challenging to state that LATS1 is enriched in non-responders

Response to reviewer's comment

We agree – this is an observational statement describing figure 2E to state that LATS1 was only detected in the NR group. To limit the risk of over interpretation a comment has been added that this paragraph and is highlighted at line 180.

Reviewer 4's comment

7. The finding that HR deficiency correlated with non-response contrast findings from other groups (Forde Nature Medicine pre0505, Minchom et al PMID 32169873).

What is the hypothesis behind this finding?

Response to reviewer's comment

Thank you for this comment. In the Forde paper, patients received both platinum doublet and ICB. HRD deficiency is associated with response to DNA damaging agents such as cisplatin and carboplatin, and so this likely explains the difference in this setting where ICB alone is being investigated. In the single patient case report by Minchom et al, the exceptional responder had a highly inflamed tumour microenvironment. This is consistent with the hypothesis that tumour inflammation (which could be shaped by the gut microbiome) is *the* dominant cause of sensitivity, compared with other genomic features in mesothelioma such as co-occurring HRD. An amendment in the discussion has been added accordingly to the discussion and highlighted at line 301.

Reviewer 4's comment

8. The independent mesothelioma cohort (MEDUSA) is introduced abruptly. The authors should clarify why this dataset was only used for some of the analyses.

Response to reviewer's comment

Thank you. The acronym MEDUSA refers to an independent study cohort which was used to enable exploration of the prognostic impact of EMT. The use of this acronym is confusing and is therefore removed from the paper.

9. The authors should include mention of the recently published UNITO-001 study of immunotherapy + PARPi in the discussion (Passiglia et al Clinical Cancer Research)

Response to reviewer's comment

This study is now included in the discussion referring to the potential for ICB+PARP. An amendment is made to the MIST5 trial adding the selection of patients with platinum sensitive mesothelioma at line 303.

Reviewer 4's comment

10. -9p21 is mentioned as encompassing CDKN2A without any mention of CDKN2B or MTAP. Recent work suggests that the immunotherapy resistance may be also be due to MTAP in other cancers (PMID 37774699, Gjuka et al)

Response to reviewer's comment

Agreed. MTAP has now been shown to regulate T cell infiltration in mesothelioma. This excellent and relevant paper has now been cited in the discussion at line 318.

Reviewer 4's comment

11. Was there any difference in gut microbiota for the 2 peritoneal cases or for patients with abdominal involvement of their cancer versus those without? Was there any difference in patients who were heavily pretreated with chemotherapy vs those who were not? This may inform timing of this regimen.

Response to reviewer's comment

Due to the limited size of this phase II study there is insufficient power to resolve with any confidence regarding differential gut microbiota composition as it relates to these two patients with peritoneal mesothelioma. It is possible that patients with primary pleural disease showing invasion may have a greater likelihood of EMT and therefore metastatic potential and resistance to ICB (based on MIST4 results), however a larger meta-analysis of ICB studies is needed to draw robust conclusions.

Reviewer 4's comment

12. The spider plot in figure 1 is blurry, as is supplementary figure 2 and supplementary table 8b

Response to reviewer's comment

Agreed, this has been replaced with a high resolution figure

Reviewer 4's comment

13. In the table, why are some patients listed as “not applicable” for p16 and BAP1 expression. Was there insufficient tissue in these cases?

Response to reviewer's comment

Correct. At line 155, the term “available archival paraffin embedded formalin fixed diagnostic tissue blocks” is added.

Reviewer 4's comment

14. For Figure 2B, the number of patients in each group should be listed under the KM curve

Response to reviewer's comment

Yes, this curve is now replotted showing the number of at risk patients corresponding under the Kaplan Meier curves. The follow up period was revised to data cut point of 4th October 2023.

REVIEWERS' COMMENTS

Reviewer #1 (Remarks to the Author):

Many thanks for your responses to our previous comments. It is easier to read and understand the manuscript now. This is an interesting study and the results could be very beneficial for the clinical practice however, as previously stated the number of participants is low to allow extrapolation. This is something that the authors have reflected more clearly in the amended manuscript. Recognising these limitations by any means does not reduce the value of the study.

Comments:

The absolute abundance of microbiota is tricky. The number of 16S rRNA genes could vary amongst bacteria. 16S rRNA Seq is a robust method but presenting the absolute number of reads might not be the best approach. For example, you mention that you started with 10ng of target DNA. DNA was extracted from stool samples. There is cross contamination with host (human) DNA. How was this issue addressed? The cross contamination of host DNA means that there is a variability of the target DNA.

Would it be possible to provide the list of total pathogens which were identified? if no negative controls were used how you filtered out the false positives?

How were bacteria classified?

In your response you mentioned that you used Illumina microarrays however, in the manuscript NGS is stated. Would it be possible to clarify the method?

Many thanks regarding your response for Figure 2C. What would happen if you remove that one patient from the NR group?

For figure 4A could you please add the bullets for each of the samples. You did this for the previous bar plots. It would be interesting to see the values of each sample.

Moreover, why these specific bacteria were chosen?

Many thanks!

Reviewer #2 (Remarks to the Author):

First of all, I'd like to thank the authors for their dedicated efforts made on this revision. Thanks to their extensive analyses and the deepened discussion, the manuscript has been very much improved. I also appreciate the change of the writing style, although it may not have been necessary at this stage of the review process. After all, I think all the concerns which I raised in the previous communication have been almost completely addressed. Again, I appreciate the careful and thorough revision by the authors. In fact, I believe this paper will be a hallmark paper. However, please further expand and deepen the clinical and basic researches on mesothelioma, so that every patient can receive an even more precise diagnosis, which should be utilized to choose his/her better therapy.

Reviewer #3 (Remarks to the Author):

The authors have appropriately responded to all my previous reviewer comments. I do not have any additional comments at this time.

Reviewer #4 (Remarks to the Author):

The revised manuscript is much improved and adequately incorporates my suggestions, as well as the comprehensive edits proposed by the other reviewers.